# On the Optimization Landscape of Low Rank Adaptation Methods for Large Language Models

**Xu-Hui Liu**[*1]    **Yali Du**[2]    **Jun Wang**[3]    **Yang Yu**[1]

[1] National Key Laboratory for Novel Software Technology, Nanjing University, Nanjing, China
[2] Cooperative AI Lab, Department of Informatics, King's College London, London, UK
[3] AI Centre, Department of Computer Science, University College London, London, UK
`liuxh@lamda.nju.edu.cn, yali.du@kcl.ac.uk, jun.wang@cs.ucl.ac.uk,`
`yuy@nju.edu.cn`

## Abstract

Training Large Language Models (LLMs) poses significant memory challenges, making low-rank adaptation methods an attractive solution. Previously, Low-Rank Adaptation (LoRA) addressed this by adding a trainable low-rank matrix to the frozen pre-trained weights in each layer, reducing the number of trainable parameters and optimizer states. GaLore, which compresses the gradient matrix instead of the weight matrix, has demonstrated superior performance to LoRA with faster convergence and reduced memory consumption. Despite their empirical success, the performance of these methods has not been fully understood or explained theoretically. In this paper, we analyze the optimization landscapes of LoRA, GaLore, and full-rank methods, revealing that GaLore benefits from fewer spurious local minima and a larger region that satisfies the $\text{PL}^*$ condition, a variant of Polyak-Łojasiewicz (PL) condition, leading to faster convergence. Our analysis leads to a novel method, GaRare, which further improves GaLore by using gradient random projection to reduce computational overhead. Practically, GaRare achieves strong performance in both pre-training and fine-tuning tasks, offering a more efficient approach to large-scale model adaptation. Code is available at `https://github.com/liuxhym/GaRare.git`.

## 1 Introduction

Large Language Models (LLMs) have demonstrated impressive performance across various domains, including natural language processing (Brown et al., 2020), computer vision (Saharia et al., 2022), and reinforcement learning (Reed et al., 2022). However, due to their large scale, pre-training and fine-tuning LLMs demand significant memory usage. Parameter-efficient fine-tuning (PEFT) techniques provide a solution by enabling the adaptation of pre-trained models to downstream applications without updating all model parameters. Among these methods, low-rank adaptation (LoRA) (Hu et al., 2022) has become a widely adopted approach. For a pre-trained weight matrix $W_0 \in \mathbb{R}^{m \times n}$, LoRA reparameterizes the fine-tuned weight matrix $W \in \mathbb{R}^{m \times n}$ as $W = W_0 + BA$, where $B \in \mathbb{R}^{m \times r}$ and $A \in \mathbb{R}^{n \times r}$. Since the rank $r$ is chosen such that $r \ll \min(m, n)$, matrices $A$ and $B$ contain significantly fewer parameters than the original matrix, resulting in reduced memory requirements during fine-tuning as $W_0$ remains fixed. Recently, Zhao et al. (2024) introduced a new method called GaLore, which compresses the rank of the gradient matrix instead of the weight matrix. This approach has been shown to outperform LoRA in terms of both performance and memory efficiency. Furthermore, GaLore is applicable to pre-training tasks, expanding its utility beyond fine-tuning alone.

To explain the success of low-rank adaptation methods, existing research has primarily focused on analyzing their expressivity (Zeng & Lee, 2024; Zhao et al., 2024) or studying the optimization landscape by assuming convex loss functions (Jang et al., 2024). However, the expressivity of low-rank

---
[*] Work done during Xu-Hui Liu's visit at King's College London.

adaptation methods alone may not fully explain the success of techniques like LoRA and GaLore, and the convexity assumption does not hold for neural networks. Notably, several studies have shown that the learning curves of different fine-tuning methods exhibit distinct patterns. For instance, Jang et al. (2024) and Xia et al. (2024) find that LoRA converges more slowly than full-rank fine-tuning, while Zhao et al. (2024) shows that GaLore converges even faster. These phenomena are tied to the optimization landscapes of these methods, which are not well-explained by current theories. The lack of a comprehensive theoretical explanation raises questions about their generalizability to other tasks and limits the development of new low-rank adaptation methods.

To address this issue, we first analyze the regions in the parameter spaces of different methods that satisfy the PL$^*$ condition (Liu et al., 2022). Under this condition, the existence of solutions is guaranteed, and the convergence rate of (S)GD is exponential. Our analysis reveals that the region in the parameter space satisfying the PL$^*$ condition condition is significantly smaller for LoRA compared to GaLore and the full-rank method. This suggests that LoRA is more likely to encounter regions of the optimization landscape where convergence guarantees are weaker, leading to slower convergence rates and suboptimal solutions. This explains why LoRA does not achieve the same convergence rate or final performance as the other two methods. To further explain GaLore's advantages over the full-rank method, we examine the presence of spurious local minima in both methods. During training, solutions can become trapped in these spurious local minima, preventing further optimization. A lower occurrence of spurious local minima implies a better optimization landscape. Our analysis shows that GaLore is more likely to meet the condition of having no spurious local minima, whereas the full-rank method is less likely to avoid them. Therefore, we conclude that GaLore offers the best optimization landscape, while LoRA presents the poorest one, which aligns with the observed empirical results. Based on our theoretical analysis, the key to preserving GaLore's favorable optimization landscape lies in projecting the gradient matrix into low-rank spaces. In GaLore, the projection is performed using Singular Value Decomposition (SVD) on the original gradient matrix, with the resulting projection matrices stored for future use. However, SVD is computationally expensive, and storing the projection matrices increases the memory usage. To enhance GaLore, we introduce **Gra**dient **Ra**ndom **Proj**ection (**GaRare**), which replaces the SVD-generated projection matrix with a randomly generated one. Theoretically, GaRare preserves the favorable optimization properties of GaLore, providing a more efficient and memory-efficient alternative while retaining similar optimization benefits.

In experiments, we first validate our theoretical findings on small datasets and MLP networks. The results demonstrate the trade-off between expressivity and the optimization landscape as predicted by the theory, as well as the similarity in the optimization processes of GaRare and GaLore. Next, we conducted experiments on LLM pre-training and fine-tuning tasks. The results show that GaRare achieves comparable performance to GaLore while using less memory, further supporting the applicability of our theoretical insights to LLMs.

Our main contributions can be summarized as follows: First, we present the first comprehensive analysis of the optimization landscapes of current mainstream fine-tuning methods: the full-rank method, LoRA, and GaLore. This theoretical framework explains the differences in convergence rates and performance among these methods. Second, building on our theoretical insights, we introduce a novel algorithm, GaRare, which retains the advantages of GaLore while further reducing memory usage. Finally, empirical results demonstrate that GaRare achieves performance comparable to GaLore in both pre-training and fine-tuning tasks, all while utilizing less memory.

## 2 BACKGROUND

### 2.1 NEURAL NETWORK

We consider input data from $D_x \subset \mathbb{R}^d$ and output data from $D_y \subset \mathbb{R}^m$. We model the data with layered, feedforward neural networks, that is, we study sets of functions $\mathcal{G} := \{g_\Theta : D_x \to \mathbb{R}^m : \Theta \in \overline{\mathcal{M}}\} \subset \mathcal{G} := \{g_\Theta : D_x \to \mathbb{R}^m : \Theta \in \mathcal{M}\}$ with

$$g_\Theta[\mathbf{x}] := \Theta^l f^l[\Theta^{l-1} \dots f^1[\Theta^0 \mathbf{x}]] \quad \text{for } \mathbf{x} \in D_x \tag{1}$$

and $\overline{\mathcal{M}} \subset \mathcal{M} := \{\Theta = (\Theta^l, \dots, \Theta^0) : \Theta^j \in \mathbb{R}^{p^{j+1} \times p^j}\}$. The quantities $p^0 = d$ and $p^{l+1} = m$ are the input and output dimensions, respectively, $l$ the depth of the networks, and $w := \min\{p^1, \dots, p^l\}$ the minimal width of the networks. The functions $f^j : \mathbb{R}^{p^j} \to \mathbb{R}^{p^j}$ are called the activation

functions. We assume that the activation functions are elementwise functions in the sense that $f^j[\mathbf{b}] = (\underline{f}^j[b_1], \ldots, \underline{f}^j[b_{p^j}])^\top$ for all $\mathbf{b} \in \mathbb{R}^{p^j}$, where $\underline{f}^j : \mathbb{R} \to \mathbb{R}$ is an arbitrary function.

Given the number of samples $n$ in the dataset, empirical-risk minimizers are the networks whose parameters are global minima of the objective function

$$\Theta \mapsto \ell(g_\Theta) := \sum_{i=1}^{n} \ell(g_\Theta[x_i], y_i) \tag{2}$$

over $\overline{\mathcal{M}}$ for fixed data $(x_1, y_1), \ldots, (x_n, y_n)$. The loss function $\ell : \mathbb{R}^m \times \mathbb{R}^m \to \mathbb{R}$ is assumed convex in its first argument; this includes all standard loss functions, such as the least-squares loss $\ell : (a, b) \mapsto \|a - b\|_2^2$, the logistic loss $\ell : (a, b) \mapsto -(1 + b)\log(1 + a) - (1 - b)\log(1 - a)$, the hinge loss $\ell : (a, b) \mapsto \max\{0, 1 - ab\}$, and so forth. We use $Dg_\Theta$ to represent the differential map of $g_\Theta \colon \mathbb{R}^t \to \mathbb{R}^m$, where $t = \sum_{j=0}^{l} p^{j+1} \times p^j$ is the number of parameters of the neural network. $Dg_\Theta$ is represented as a $m \times t$ matrix, with $(Dg_\Theta)_{i,j} := \frac{\partial(g_\Theta)_i}{\partial \Theta_j}$. The neural tangent kernel (Jacot et al., 2018), is defined as an $m \times m$ matrix $K(\Theta) = Dg_\Theta(Dg_\Theta)^T$.

## 2.2 GALORE

Gradient low-rank projection (GaLore) (Zhao et al., 2024) demonstrate the following gradient update rules: $\Theta_T = \Theta_0 + \eta \sum_{t=0}^{T-1} \Delta G_t$, and $\Delta G_t = \hat{U}_t \rho_t(\hat{U}_t^T G_t)$, where $\eta$ is the learning rate, $G_t$ is the gradient matrices, $\hat{U}_t$ is the projection matrices, $\rho_t$ is an entry-wise stateful gradient regularizer. For example, for Adam, $\rho_t(G_t) = M_t / \sqrt{V_t + \epsilon}$, where $M_t = \beta_1 M_{t-1} + (1 - \beta_1)G_t$, and $V_t = \beta_2 V_{t-1} + (1 - \beta_2)G_t^2$. Therefore, $\rho_t$ can be memory-intensive. Projecting gradient matrix to a low-rank space can reduce the memory usage significantly. In GaLore, the projection matrices is selected based on Singular Value Decomposition (SVD): $G_t = U\Sigma V^T$, $\hat{U}_t = U[:, : r]$ is the first $r$ columns of $U$.

## 3 RELATED WORK

**Low-rank adaptation.** Hu et al. (2022) proposed Low-Rank Adaptation (LoRA) to fine-tune pre-trained models with reduced memory usage. Building on LoRA, numerous methods have been introduced to enhance its performance (Liu et al., 2024; Yang et al., 2024; Lin et al., 2024a;b; Hayou et al., 2024; Lialin et al., 2024; Meng et al., 2024). For example, LoRA Dropout(Lin et al., 2024b) introduces random noise into the learnable low-rank matrices, increasing parameter sparsity to reduce overfitting. LoRA+(Hayou et al., 2024) assigns different learning rates to the low-rank matrices to improve training efficiency, while PiZZA (Meng et al., 2024) employs Singular Value Decomposition (SVD) on weight matrices to extract principal components as low-rank matrices.

However, the optimization landscape of the LoRA-based framework restricts its application to fine-tuning tasks only. ReLoRA(Lialin et al., 2024) extends LoRA for pre-training but requires a full-rank warm-up phase to match the performance of standard pre-training. FLoRA, by establishing an equivalence between LoRA and gradient compression, eliminates the need for a warm-up phase in pre-training. GaLore(Zhao et al., 2024) projects gradient matrices into low-rank spaces, achieving strong performance and high memory efficiency in both pre-training and fine-tuning tasks. Building on GaLore, WeLore(Jaiswal et al., 2024) adaptively selects the rank of projection matrices, and OwLore(Li et al., 2024) introduces layer-wise updates to further improve flexibility and efficiency.

**Theory of neural networks.** The expressivity of neural networks refers to their ability to approximate a target function. This concept is foundational, starting with the universal approximation theorems (Cybenko, 1989; Hornik et al., 1990), which demonstrate that neural networks, under certain conditions, can approximate any continuous function. Building on this, a series of subsequent works further explored the limitations and potential of neural networks (Poon & Domingos, 2011; Bengio & Delalleau, 2011; Lu et al., 2017; Li et al., 2023). From a different angle, studies on the loss landscape (Du et al., 2019; Zou et al., 2018; Allen-Zhu et al., 2019; Liu et al., 2022) have provided insights into how first-order optimization methods, such as gradient descent, are able to converge to global minima under certain conditions.

**Theory of low-rank adaptation.** Aghajanyan et al. (2021) found that an intrinsic low-rank structure is crucial for fine-tuning language models. Zeng & Lee (2024) analyzed the expressive power of LoRA, and Zhao et al. (2024) examined the low-rank structure of gradient matrices. While these studies conclude that low-rank adaptation methods can achieve strong performance, the details of the optimization process remain unclear. Jang et al. (2024) attempted to analyze the optimization landscape of LoRA, but their analysis relies on the convexity assumption.

## 4  ANALYSIS

In this section, we analyze the optimization landscape of GaLore, LoRA and full-rank method. The analysis is divided into two parts. The first part found the optimization landscapes of GaLore and full-rank method are superior to that of LoRA. This part is based on PL$^*$ condition (Oymak & Soltanolkotabi, 2020), a variant of Polyak-Łojasiewicz (PL) condition, which is a framework for analyzing non-convexity systems, such as neural networks. PL$^*$ condition is important because the loss function that satisfies PL$^*$ condition is guaranteed with exponential convergence rate under gradient descent (GD) and stochastic gradient descent (SGD). Therefore, the larger the region satisfied PL$^*$ condition, the better the optimization landscape is. Liu et al. (2022) found that over-parameterized systems have high probability to satisfies PL$^*$ condition, which explains why large models always demonstrate better performance (Brown et al., 2020). Intuitively, LoRA harms the over-parameterization of the neural network, as it only optimizes much smaller number of parameters than full-rank method. In contrast, GaLore retains the over-parameterization by optimizing all parameters. We verify this intuition by calculating the region where the PL$^*$ condition is satisfied for all three methods.

The second part finds GaLore has a better optimization landscape than full-rank method. This analysis focuses on spurious local minima—fewer spurious local minima imply a better optimization landscape, as the optimization process is less likely to get trapped in poor solutions. We find that it is challenging for the full-rank method to avoid spurious local minima, whereas GaLore easily achieves this. Due to space constraints, the proofs for this section are provided in Appendix A.

### 4.1  GRADIENT LOW-RANK PROJECTION KEEPS OVER-PARAMETERIZATION

To show the relationship between over-parameterization and good optimization landscape, we first introduce the definition of PL$^*$ condition: A non-negative function $\mathcal{L}$ satisfies $\mu$-PL$^*$ condition on a set $\mathcal{M} \in \mathbb{R}^t$ for $\mu > 0$, if $\|\nabla\mathcal{L}(\Theta)\|^2 \geq \mu\mathcal{L}(\Theta)$, $\forall \Theta \in \mathcal{M}$. As we mentioned earlier, PL$^*$ condition ensures the existence of solutions and guarantees exponential convergence for GD and SGD, implying a good optimization landscape. In this section, we analyze the regions where the PL$^*$ condition is satisfied for each method. The following theorem reveals the relationship between PL$^*$ condition and over-parameterization. The connection is established by the eigenvalues of the neural target kernel $K(\Theta)$. Let $\lambda_{\min}(K(\Theta))$ be the smallest eigenvalue of $K(\Theta)$, we have

**Theorem 4.1** (Theorem 1 and Proposition 4 of (Liu et al., 2022)). *The square loss function $\ell(g_\Theta) = \frac{1}{2}\|g_\Theta(\mathbf{x}) - \mathbf{y}\|^2$ satisfies $\mu$-PL$^*$ condition on $\mathcal{M}$ if and only if $\lambda_{\min}(K(\Theta)) \geq \mu > 0$.*

$\lambda_{\min}(K(\Theta)) > 0$ implies $K(\Theta)$ is not singular. PL$^*$ condition holds across most of the parameter space for over-parameterized systems, the intuition behind it is based on parameter counting (Liu et al., 2022). Note that $K(\Theta) = Dg_\Theta(Dg_\Theta)^T$, the singular set of $\Theta$, such that $K(\Theta)$ is not full rank will of co-dimension $t - n + 1$, where $t$ is the number of parameters, $n$ is the number of samples. If $t < n$, i.e., the systems are under-parameterized, $K(\Theta)$ is always rank deficient and therefore $\lambda_{\min}(K(\Theta)) = 0$. Hence such systems never satisfy PL$^*$ condition. On the other side, the larger the degree of the model over-parameterization $t - n$ is, the smaller the singular set is expected to be. This intuition explains why LoRA shows worse performance compared to full rank finetuning and Galore: LoRA reduces the number of parameters of each layer, the degree of over-parameterization is reduced and even the systems become under-parameterized. Therefore, the PL$^*$ condition is difficult to be satisfied if LoRA is used. In contrast, although Galore projects the gradient matrices to the low-rank spaces, the projected matrices will be projected back during the parameters updating process, and thus the degree if over-parameterization of the original systems are kept. The theoretical guarantee for full-rank training is established by Liu et al. (2022).

**Theorem 4.2** (Theorem 4 of Liu et al. (2022)). *Consider the neural network defined by Eq. (1), where the initial parameter $\Theta_0$ is randomly chosen such that $\Theta_0^l \sim \mathcal{N}(0, I_{p^j,p^{j+1}})$ for $j \in [l+1]$.*

*Let $\lambda_0 := \lambda_{\min}(K(\Theta_0)) > 0$ denote the minimum eigenvalue of the kernel matrix $K(\Theta_0)$. For any $\mu \in (0, \lambda_0)$, if the width of the network satisfies $w = \tilde{\Omega}\left(\frac{mnR^{6l+2}}{(\lambda_0-\mu)^2}\right)$, then the square loss satisfies $\mu$-PL\* condition with high probability over the ball $B(\Theta_0, R)$ for full-rank method.*

According to this theorem, the radius of the region that satisfies the PL\* condition depends on the width of the network. The wider the network, the larger this region becomes, confirming the intuition that over-parameterization leads to the PL\* condition and thus a favorable optimization landscape. Based on the derivation of Theorem 4.2, we extend the results to GaLore and LoRA, respectively.

**Theorem 4.3.** *Under the condition of Theorem 4.2, let $r$ be the rank of GaLore, if the number of training samples $n$ satisfies $n \le r \sum_{j=1}^{l} p^j$, then if the width of the network $w = \tilde{\Omega}\left(\frac{mnR^{6l+2}}{(\lambda_0-\mu)^2}\right)$, the square loss satisfies $\mu$-PL\* condition with high probability over the ball $B(\Theta_0, R)$ for GaLore.*

This theorem demonstrates that the GaLore method covers the same region as the full-rank method, with the exception of an additional mild condition: the number of training samples $n \le r \sum_{j=1}^{l} p^j$. This condition is easy to satisfy. For instance, in the case of a 1B model, if the rank is selected as $r = 512$, the constraint on the number of training tokens becomes 512 billion. This constraint exceeds the number of tokens required to train a 1B model. Then we present the result for LoRA.

**Theorem 4.4.** *Under the condition of Theorem 4.2, let $r$ be the rank of LoRA, if $r = \tilde{\Omega}\left(\frac{mnR^{12l+2}}{(\lambda_0-\mu)^2}\right)$, then the square loss satisfies $\mu$-PL\* condition with high probability over the ball $B(\Theta_0, R)$ for LoRA.*

The theoretical results support our intuition: the region satisfying PL\* condition is significantly smaller in LoRA compared to the Galore and full-rank methods. This is evident from the fact that $r \ll w$ and the exponent of $R$ increases from $6l + 2$ to $12l + 2$, resulting in a much smaller radius $R$ under a fixed $r$. This reduced region indicates that the space where a favorable optimization landscape exists — characterized by the presence of solutions and rapid convergence rates — is more constrained in LoRA. Consequently, LoRA exhibits poorer performance compared to the others.

## 4.2 Gradient Low-Rank Projection Has No Spurious Minima

Spurious local minima are entities that optimization algorithms strive to avoid. Intuitively, a spurious local minimum is a point in the parameter space where the loss in its neighbourhood is higher than at the point itself, thereby preventing the optimization algorithm from escaping from this point. Formally, we use the definition of spurious local minima in Lederer (2020).

**Definition 4.5** (Spurious local minima). Let $\Theta \in \overline{\mathcal{M}}$ be a local minimum of the objective function (2). If there is no continuous function $h : [0, 1] \to \overline{\mathcal{M}}$ that satisfies (i) $h[0] = \Theta$ and $h[1] = \Gamma$ for a global minimum $\Gamma \in \overline{\mathcal{M}}$ of the objective function (2) and (ii) $t \mapsto \ell(g_h[t])$ is nonincreasing, we call the parameter $\Theta$ a spurious local minimum.

According to this definition, if there is no spurious local minima in $\overline{\mathcal{M}}$, then there exists at least one non-increasing path belongs to $\overline{\mathcal{M}}$ from every point in $\overline{\mathcal{M}}$ to the global minima. For full-rank method, the optimized parameter space $\overline{\mathcal{M}}$ is exactly the original parameter space $\mathcal{M}$. For the whole parameter space, Lederer (2020) gives the result of full-rank method.

**Theorem 4.6** (Theorem 1 of Lederer (2020)). *For the network defined by Eq. (1), if $w$, the minimal width of the network, no less than $2m(n + 1)^l$, then the objective function (2) has no spurious local minima with full-rank method.*

This theorem demonstrates that neural network has no spurious local minima if it is wide enough. However, the minimal width $2m(n + 1)^l$ is large and cannot be satisfied in application, especially for LMs, where $n$ and $l$ are much larger.

The result can be generalized to LoRA easily by considering the network of LoRA as

$$g_\Theta[\mathbf{x}] := \Theta_2^l I[\Theta_1^l f^l[\Theta_2^{l-1} I[\Theta_1^{l-1} \dots f^1[\Theta_2^0 I[\Theta_1^0 \mathbf{x}]]], \tag{3}$$

where $\Theta_2^j \in \mathbb{R}^{p^j \times r}$, $\Theta_1^j \in \mathbb{R}^{r \times p^{j-1}}$, $r$ is the rank for LoRA, and $I$ is the identity function.

**Corollary 4.7** (No spurious local minima condition for LoRA). *For the network defined by Eq. (3), if $r$, the rank of LoRA, no less than $2m(n + 1)^{2l}$, then the objective function (2) has no spurious local minima with LoRA.*

It can be seen that the condition for LoRA is even more difficult to satisfy. The facts that $r \ll w$ and the exponent from $l$ to $2l$ implies LoRA cannot guarantee no spurious local minima in the optimization process.

For GaLore, while updates are performed within the low-rank parameter space $\overline{\mathcal{M}}$, the gradient computations are carried out in the original full-rank space $\mathcal{M}$. Consequently, whether a point is a spurious local minima of GaLore is determined by the optimization landscape in $\mathcal{M}$, rather than in $\overline{\mathcal{M}}$. However, this does not imply that the optimization landscape of GaLore is identical to that of the full-rank method. This difference arises because GaLore constrains the parameters to remain within $\overline{\mathcal{M}}$ after each projection. Thus, the optimization process is restricted to $\overline{\mathcal{M}}$, while the determination of spurious local minima depends on the landscape in $\mathcal{M}$. To formalize this concept, we introduce the definition of projected spurious local minima.

**Definition 4.8** (Projected spurious local minima). Let $\Theta \in \overline{\mathcal{M}}$ be a local minimum of the objective function (2). If there is no continuous function $h : [0,1] \to \mathcal{M}$ that satisfies (i) $h[0] = \Theta$ and $h[1] = \Gamma$ for a global minimum $\Gamma \in \overline{\mathcal{M}}$ of the objective function (2) and (ii) $t \mapsto \ell(g_h[t])$ is nonincreasing, we call the parameter $\Theta$ a projected spurious local minimum.

Based on the definition, we have the condition of no spurious local minima for GaLore.

**Theorem 4.9** (No projected spurious local minima condition for GaLore). *For the network defined by Eq. (1), if $w$, minimal width of the network, no less than $2r$, where $r$ is the rank of GaLore, then the objective function (2) has no projected spurious local minima with GaLore.*

The condition is $2r$ instead of $2m(n+1)^l$ indicates that GaLore is less vulnerable to spurious local minima compared to the full-rank method. Spurious local minima are suboptimal solutions that can trap the optimizer and prevent it from reaching a globally or near-globally optimal solution. By reducing the likelihood of encountering such minima, GaLore benefits from a smoother and more favorable optimization landscape, which allows the optimizer to explore more promising regions of the parameter space. This explains why GaLore trains faster and, in some cases, surpasses the full-rank method in performance. This theorem is further verified in Figure 2 of Section 6.1.

### 4.3 DISCUSSION

In the above, we establish that GaLore offers the most favorable optimization landscape, whereas LoRA's landscape is the least advantageous. However, this does not necessarily mean GaLore will outperform the full-rank method in terms of final results. The superiority of one method depends on whether expressiveness or the optimization landscape's quality has a greater impact. Intuitively, for smaller networks, where the optimization process is simpler but expressivity is limited, the full-rank method may perform better. Conversely, for larger networks, where expressivity is sufficient for good results and the optimization process is more complex, GaLore tends to excel. According to Theorem 4.9, the absence of spurious minima is guaranteed only when the rank $r$ is small. Initially, increasing $r$ improves GaLore's expressivity, allowing it to model more complex functions effectively. However, if $r$ becomes too large, the optimization landscape may degrade due to the emergence of spurious local minima or poorly conditioned regions, even though expressivity continues to increase. When expressivity is not the primary performance bottleneck, we expect GaLore's performance to improve with increasing $r$, but decline as $r$ approaches the network's width, where the landscape quality diminishes. These conclusions are tested and validated in Section 6.1, with empirical results confirming this theoretical behavior.

## 5 GARARE: GRADIENT RANDOM PROJECTION

According to the analysis in Section 4, GaLore has good optimization landscape, characterized by the same range of region that satisfy the PL$^*$ condition condition and a higher likelihood of avoiding spurious local minima compared to full-rank methods. The key point is GaLore project the gradient matrix to low-rank spaces with projection matrices. In GaLore the matrices are obtained by performing SVD to the gradient matrix, which is both time-consuming and memory-consuming. However, to achieve the low-rank projection, we only need matrices with the same size as those of GaLore. Inspired by this intuition, we propose our new algorithm, which uses random matrices instead of the matrices derived by SVD.

Table 1: Comparison of GaRare, GaLore and LoRA. Assume $\Theta \in \mathbb{R}^{p \times q}$, rank $r$.

|  | GaRare | GaLore | LoRA |
|---|---|---|---|
| Weights | $pq$ | $pq$ | $pq + pr + qr$ |
| Optim States | $2qr$ | $pr + 2qr$ | $2pr + 2qr$ |
| Multi-Subspace | ✓ | ✓ | × |
| Pre-Training | ✓ | ✓ | × |
| Fine-Tuning | ✓ | ✓ | ✓ |

## 5.1 ALGORITHM AND THEORETICAL GUARANTEE

Note that it is infeasible to use arbitrary matrices that have the same size as GaLore, the vectors contained within these matrices must be linearly independent; otherwise, the projected spaces will have an effective rank lower than $r$. Fortunately, the following lemma shows the matrix generated by Gaussian distribution satisfies this almost surely.

**Lemma 5.1.** *Given a matrix $R \in \mathbb{R}^{p \times r}$ ($r < m$), the elements of $R$ are generated by Gaussian distribution $\mathcal{N}(0, \sigma^2)$, then we have $P(\mathrm{rank}(R) = r) = 1$.*

The proof is deferred to Appendix A.3. Based on this result, we analyze whether the random matrix can maintain the property identified by Theorem 4.3 and 4.9.

**Corollary 5.2.** *Under the condition of Theorem 4.2, if the random matrix $R \in \mathbb{R}^{p \times r}$ is generated from $\mathcal{N}(0, 1/p)$, and the number of training samples $n \leq r \sum_{j=1}^{l} p^j$, then if the network width $w = \tilde{\Omega}\left(\frac{mnR^{6l+2}}{(\lambda_0 - \mu)^2}\right)$, the square loss satisfies $\mu$-$PL^*$ condition with high probability over the ball $B(\Theta_0, R)$.*

**Corollary 5.3.** *For the network defined by Eq. (1), if $w$, minimal width of the network, no less than $2r$, where $r$ is the smaller dimension of the projection matrix, then the objective function (2) has no projected spurious local minima.*

The proof of the two corollaries is deferred to Appendix A.3. These corollaries ensure that GaLore's properties are preserved as long as the projection matrix is generated using $\mathcal{N}(0, 1/p)$. Based on this, we propose **Gra**dient **Ra**ndom P**r**oj**e**ction (**GaRare**), which replaces the projection matrices in GaLore with certain random matrices. GaRare retains the same hyperparameters as GaLore, including the scale factor $\alpha$ and subspace update frequency $T$. The scale factor $\alpha$ controls the strength of the low-rank update, similar to the $\alpha/r$ scale factor used in the low-rank adaptor of Hu et al. (2022). The subspace update frequency $T$ specifies how often the random matrices are refreshed, with the random seed changing every $T$ steps. The pseudo-code for GaRare is provided in Appendix C.

The process of GaRare seems similar to that of FLoRA (Hao et al., 2024), as both use a random matrix to project gradients. However, FLoRA is derived from the theory of LoRA and is designed to mimic its learning dynamics. In FLoRA, the random matrix is generated using $\mathcal{N}(0, 1/pr)$, which leads to a violation of Corollary 5.2. Consequently, FLoRA does not share the same optimization landscape as GaLore and GaRare. This distinction is further verified in our experimental results.

## 5.2 MEMORY USAGE OF GARARE

We briefly analyze the memory usage of LoRA, GaLore, and GaRare here, with a more detailed discussion in Appendix G. LoRA requires storing the low-rank matrices $B \in \mathbb{R}^{p \times r}$ and $A \in \mathbb{R}^{r \times q}$, along with the full-rank weight matrix $\Theta \in \mathbb{R}^{p \times q}$, resulting in $pq + pr + qr$ weights, plus an additional $2pr + 2qr$ for optimizer states. GaLore reduces storage by eliminating $B$ and $A$, retaining $pq$ weights but adding $pr + 2qr$ for optimizer states and the projection matrix. GaRare uses the same structure as GaLore but further minimizes memory usage by dynamically generating a random matrix at each iteration, requiring only $2qr$ for optimizer states and avoiding storing the projection matrix. A summary of memory usage is provided in Table 1. Although FLoRA matches GaRare in memory usage, its inferior optimization performance makes GaRare the focus of this analysis.

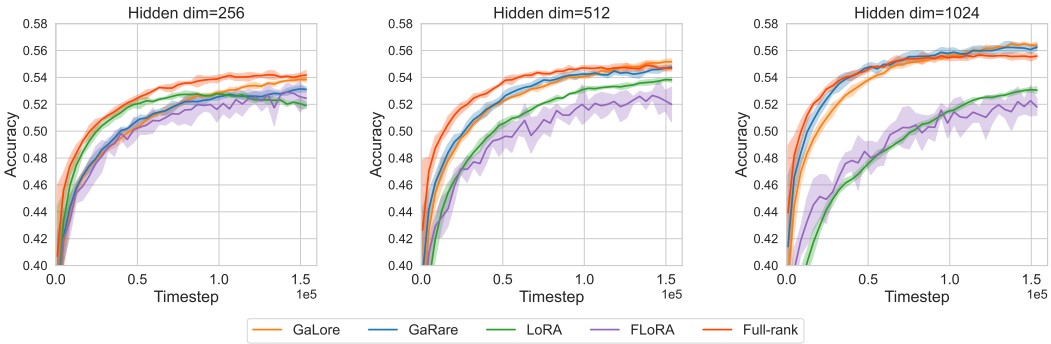

Figure 1: The performance of GaLore, GaRare, LoRA and Full-rank method on CIFAR-10 dataset. The dimension of hidden layers is set to be 256, 512 and 1024, while the rank for the low-rank adaptation methods are set as 128, 256 and 512, respectively.

## 6 EXPERIMENTS

In this section, we first verify our proposed theorems in a simple dataset. Then we evaluation GaRare on both pre-training and fine-tuning of LLMs. All experiments run on NVIDIA A100 GPUs.

### 6.1 VERIFICATION OF THE THEORY

In this section, we describe the experiments conducted to validate our theoretical claims related to feedforward neural networks, specifically Multi-Layer Perceptrons (MLPs). We use the CIFAR-10 dataset (Torralba et al., 2008) for our experiments. All experiments are training in 30 epochs. The learning rate is selected from a set of $\{1e^{-2}, 1e^{-3}, \ldots, 1e^{-6}\}$, and the best learning rate is chosen based on the final accuracy. A detailed description of the network architecture is provided in Appendix B.1. The experiment results are averaged over three random seeds.

As discussed in Section 4.3, Theorems 4.3 and 4.9 indicate a trade-off between the expressivity of a network and its optimization landscape in GaLore. To illustrate this, we adjust the hidden layer dimensions in the MLPs to three settings: 256, 512, and 1024. When the dimension is smaller, expressivity limits performance; conversely, larger dimensions improve optimization landscapes, leading to enhanced performance. Additionally, GaRare is expected to perform similarly to GaLore, as indicated by Corollaries 5.2 and 5.3. In our experiments, we set the rank for low-rank adaptation methods to half of the hidden layer dimensions, specifically 128, 256, and 512. The primary metric for evaluation is the prediction accuracy on the test dataset.

The experimental results, illustrated in Figure 1, show that LoRA exhibits worse performance than GaLore and GaRare across the three network configurations, underscoring its suboptimal optimization landscape. FLoRA shows similar performance as LoRA, which in line with the conclusion of Hao et al. (2024). In contrast, the performance of the other three methods is better especially when the dimensionality of the hidden layers is large. Specifically, with a hidden dimension of 256, the full-rank method out-

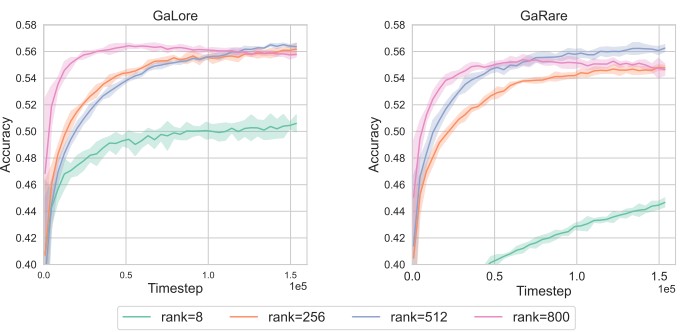

Figure 2: The performance of GaLore and GaRare on CIFAR-10 dataset with different ranks.

performs GaLore and GaRare due to the limited expressivity being the performance bottleneck. At a hidden dimension of 512, all three methods demonstrate similar performance levels. However, when the hidden dimension increases to 1024, both GaLore and GaRare outperform the full-rank

Table 3: Comparison with low-rank adaptation methods on pre-training various sizes of LLaMA models on C4 dataset. Validation perplexity is reported, along with a memory estimate of the total of parameters and optimizer states based on BF16 format.

|  | 60M | 130M | 350M | 1B |
|---|---|---|---|---|
| Full-Rank | 34.06 (0.34G) | 25.08 (0.79G) | 18.80 (2.16G) | 15.56 (7.85G) |
| LoRA | 34.99 (0.28G) | 33.92 (0.63G) | 25.58 (1.38G) | 19.21 (4.75G) |
| FLoRA | 36.97 (0.26G) | 30.22 (0.53G) | 22.67 (1.19G) | 20.22 (3.98G) |
| GaLore | 34.88 (0.26G) | 25.36 (0.56G) | **18.95** (1.25G) | **15.64** (4.23G) |
| GaRare | **34.45** (0.26G) | **25.34** (0.53G) | 19.35 (1.19G) | 15.88 (3.98G) |
| $r / d_{\text{model}}$ | 128 / 256 | 256 / 768 | 256 / 1024 | 512 / 2048 |
| Training Tokens | 1.1B | 2.2B | 6.4B | 13.1B |

method, suggesting that sufficient expressivity allows the optimization landscape to dominate the final performance outcomes.

Furthermore, we explore the impact of rank selection on GaLore and GaRare by setting the hidden dimension at 1024 and testing four different ranks: 8, 256, 512, and 800. According to Theorem 4.9, the absence of spurious local minima is guaranteed when the rank does not exceed 512. Therefore, we hypothesize that performance should increase as the rank increases up to 512, due to improved expressivity without compromising the optimization landscape. Beyond this threshold, however, a trade-off between expressivity and optimization landscape might impair performance. The results, depicted in Figure 2, support this hypothesis: performance improves as rank increases from 8 to 512, but begins to decline when the rank reaches 800.

It is also noteworthy that GaLore and GaRare exhibit similar performance in most experimental conditions, with exceptions when the hidden dimension is 256 with a rank of 128, and when the hidden dimension is 1024 with a rank of 8. These deviations can be attributed to Cor. 5.2 and 5.3, which apply when the network is sufficiently large. The discrepancies in these specific conditions suggest that the network scale is too small to align with these theoretical assurances. These experiments on the CIFAR-10 dataset validate our theoretical deductions and confirm the intricate dynamics between network configuration and performance outcomes.

## 6.2 PERFORMANCE ON PRE-TRAINING TASKS

To evaluate the performance of GaRare, we apply it to train LLaMA-based large language models using the C4 dataset. The C4 dataset is a colossal, cleaned version of the Common Crawl web corpus, designed primarily for pre-training language models and word representations (Raffel et al., 2019). We follow the experimental setup outlined by Zhao et al. (2024), which utilizes a LLaMA-based architecture incorporating RMSNorm and SwiGLU activations (Shazeer, 2020; Touvron et al., 2023). For each model size, we use the same set of hyperparameters across methods, except for the learning rate. Both GaRare and FLoRA have their learning rates tuned under the same computational budget, and we report the best performance achieved. For the other methods, we present the results reported in Zhao et al. (2024). All experiments are conducted in BF16 format to optimize memory usage. Detailed descriptions of our task setups and hyperparameters are provided in Appendix B.2.

Table 2: Pre-training LLaMA 7B on C4 dataset for 30K steps. Validation perplexity and memory estimate are reported.

|  | Mem | 30K |
|---|---|---|
| 8-bit GaRare | **16G** | 19.94 |
| 8-bit GaLore | 18G | 19.93 |
| 8-bit Adam | 26G | 20.05 |
| **Tokens (B)** |  | 3.9 |

We report the final validation perplexity of the methods in Table 3. GaRare, GaLore, and the full-rank method exhibit comparable performance, outperforming other low-rank adaptation methods. While GaLore reduces memory usage, GaRare achieves even greater memory efficiency. Although FLoRA matches GaRare in memory usage, its performance is significantly worse. Furthermore, the comparable performance of GaLore and GaRare in the LLaMA-based structure suggests that our theory has the ability to extend to other network architectures.

To demonstrate the scalability of GaRare, we train the 7B model using this method. Similar to Ga-Lore, GaRare can be applied to various learning algorithms, particularly memory-efficient optimizers, to further reduce the memory footprint. In this experiment, we incorporate the 8-bit Adam technique (Loshchilov & Hutter, 2019) and compare 8-bit GaRare with 8-bit GaLore and 8-bit Adam. Due to computational resource constraints, the model is trained for 30K steps using 3.9B tokens. The results are presented in Table 2. GaRare achieves performance comparable to GaLore while providing greater memory efficiency, reducing memory usage by up to 10%, and eliminating the need for SVD computation. Notably, GaRare and GaLore outperform the full-rank method within the 30K steps, highlighting the superiority of their optimization landscapes.

## 6.3 PERFORMANCE ON FINE-TUNING TASKS

We evaluate GaRare for fine-tuning on the GLUE benchmark (Wang et al., 2019), which comprises a variety of tasks such as sentiment analysis, question answering, and textual entailment. In this experiment, we fine-tune pre-trained RoBERTa models on GLUE tasks using GaRare and compare its performance with other methods. For LoRA, we use the hyperparameters from Hu et al. (2022), for GaLore, we use the settings from Zhao et al. (2024), and for GaRare and FLoRA, we tune the learning rate and batch size. Detailed hyperparameters can be found in Appendix B.3. As shown in Table 4, GaRare delivers performance comparable to GaLore while outperforming other low-rank adaptation methods in both RoBERTa-Base and RoBERTa-Large models. GaRare has the smallest memory footprint among all methods. Although FLoRA has the same memory usage as GaRare, it exhibits poorer performance. This demonstrates that GaRare is an effective full-stack memory-efficient training strategy for both LLM pre-training and fine-tuning.

Table 4: Performance on GLUE with RoBERTa-Base and RoBERTa-Large model. We report the average score of all tasks.

| | RoBERTa-Base | | | | | RoBERTa-Large | | | | |
|---|---|---|---|---|---|---|---|---|---|---|
| | **Full-Rank** | **GaRare** | **GaLore** | **FLoRA** | **LoRA** | **Full-Rank** | **GaRare** | **GaLore** | **FLoRA** | **LoRA** |
| Memory | 748M | 252M | 253M | 252M | 257M | 2132M | 718M | 720M | 718M | 732M |
| **CoLA** | 62.2 | 61.1 | 60.4 | 59.0 | **61.4** | 68.0 | 67.9 | **68.3** | 65.5 | 68.2 |
| **STS-B** | 90.9 | 90.3 | **90.7** | 89.9 | 90.6 | 91.5 | 92.3 | 92.5 | 92.5 | **92.6** |
| **MRPC** | 91.3 | 91.5 | **92.3** | 88.5 | 91.1 | 90.9 | **91.7** | **91.7** | 89.3 | 90.9 |
| **RTE** | 79.4 | 79.3 | **79.4** | 76.5 | 78.7 | 86.6 | **87.4** | 87.0 | 83.0 | **87.4** |
| **SST2** | 94.6 | **94.4** | 94.0 | 93.8 | 92.9 | 96.4 | **96.2** | 96.1 | 96.0 | **96.2** |
| **MNLI** | 87.2 | **87.2** | 87.0 | 86.6 | 86.8 | 90.2 | **91.3** | 90.8 | 90.4 | 90.6 |
| **QNLI** | 92.3 | **92.3** | 92.2 | 91.9 | 92.2 | 94.7 | 94.6 | **95.7** | 94.0 | 94.9 |
| **QQP** | 92.3 | 90.9 | 91.1 | 90.9 | **91.3** | 92.2 | 91.8 | **91.9** | 91.5 | 91.5 |
| **Avg** | 86.3 | **85.9** | **85.9** | 84.6 | 85.6 | 88.8 | 89.2 | **89.3** | 87.8 | 89.0 |

## 7 CONCLUSION

This work presents a comprehensive analysis of the optimization landscapes of prevalent fine-tuning methods, including the full-rank approach, LoRA, and GaLore. Our theoretical findings reveal that GaLore offers a more favorable optimization landscape compared to LoRA and the full-rank method, as it is less prone to spurious local minima and operates within a broader region that satisfies the $PL^*$ condition. This explains GaLore's faster convergence and superior performance relative to the other methods. In contrast, LoRA, while being memory-efficient, struggles with a more challenging optimization landscape, which accounts for its slower convergence. Based on these insights, we introduce GaRare, a memory-efficient alternative to GaLore that retains similar optimization benefits. Experimental results confirm GaRare's effectiveness in large language model tasks. Additionally, our theoretical findings highlight the power of the gradient projection framework, underscoring its potential to drive future advancements in efficient fine-tuning algorithms.

**Limitations:** Our theoretical analysis is confined to MLP architectures, which limits the applicability of our findings and may also impact the performance of the derived algorithms. We consider extending this analysis to other architectures, such as Transformers, as part of our future work. New theoretical insights could lead to the development of novel algorithms, potentially resulting in further performance enhancements and reduced memory usage.

## 8 REPRODUCIBILITY STATEMENT

The proof of the theoretical results is provided in Appendix A. The assumptions and conditions for the theorems are stated within the theorems themselves, and limitations are discussed in Sections 4.3 and 7.

The pseudo-code for the algorithm is available in Appendix C. The model structure and hyperparameters are detailed in Appendix B. The source code is included in the supplementary material.

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

## A   DETAILS OF THE PROOFS

### A.1   PROOFS OF SECTION 4.1

#### A.1.1   DERIVATION OF THEOREM 4.1

Theorem 4.1 comes from two results from Liu et al. (2022).

**Theorem A.1** (Theorem 1 of Liu et al. (2022))**.** *If $\lambda_{\min}(K(\Theta)) \geq \mu > 0$, $\forall \Theta \in \mathcal{M}$, then the square loss function $\ell(g_\Theta) = \frac{1}{2}\|g_\Theta(\mathbf{x}) - \mathbf{y}\|^2$ satisfies $\mu$-$PL^*$ condition on $\mathcal{M}$.*

*Proof.*

$$\frac{1}{2}\|\nabla\ell(g_\Theta)\|^2 = \frac{1}{2}(g_\Theta(\mathbf{x}) - \mathbf{y})^T K(\Theta)(g_\Theta(\mathbf{x}) - \mathbf{y})$$

$$\geq \frac{1}{2}\lambda_{\min}(K(\Theta))\|g_\Theta(\mathbf{x}) - \mathbf{y}\|^2 = \lambda_{\min}(K(\Theta))\ell(g_\Theta) \geq \mu\ell(g_\Theta)$$

□

This theorem shows $\lambda_{\min}(K(\Theta)) \geq \mu > 0$ is a sufficient condition of PL* condition. It is also a necessary condition in a certain sense:

**Theorem A.2** (Proposition 4 of (Liu et al., 2022)). *If $\lambda_{\min}(K(\Theta_0)) = 0$, then the system $g_\Theta(\mathbf{x}) = \mathbf{y}$ cannot be PL* condition for all $\mathbf{y}$ on any set $\mathcal{M}$ that contains $\Theta_0$.*

*Proof.* Since $\lambda_{\min}(K(\Theta_0)) = 0$, we can choose $rvy$ so that $K(\Theta_0)(g_{\Theta_0}(\mathbf{x}) - \mathbf{y}) = 0$ and $g_{\Theta_0}(\mathbf{x}) - \mathbf{y} \neq 0$. Therefore,

$$\frac{1}{2}\|\nabla\ell(g_{\Theta_0})\|^2 = \frac{1}{2}(g_\Theta(\mathbf{x}) - \mathbf{y})^T K(\Theta)(g_\Theta(\mathbf{x}) - \mathbf{y}) = 0.$$

□

This theorem implies it is also a necessary condition.

### A.1.2 PROOF OF THEOREM 4.3

We first introduce one definition and two useful lemmas.

**Definition A.3** (Uniform conditioning). We say that $g_\Theta$ is $\mu$-uniformly conditioned ($\mu > 0$) on $\mathcal{M} \subset \mathbb{R}^t$ if the smallest eigenvalue of its targent kernel $K(\Theta)$ satisfies

$$\lambda_{\min}(K(\Theta)) \geq \mu, \forall\Theta \in \mathcal{M}.$$

According to Theorem A.1, if $K(\Theta)$ is $\mu$-uniformly conditioned, it is $\mu$-PL* condition. Let $H_g$ be the Hessian matrix of function $g$, i.e., $(H_g)_{ij} = \frac{\partial^2 g_i}{\partial\Theta_i\partial\Theta_j}$.

**Lemma A.4** (Theorem 2 of Liu et al. (2022)). *Given $\Theta_0$, suppose the targent kernel matrix $K(\Theta_0)$ is strictly positive definite, i.e., $\lambda_0 := \lambda_{\min}(K(\Theta_0)) > 0$. If the Hessian spectral norm $\|H_g\| \leq \frac{\lambda_0 - \mu}{2L_g\sqrt{n}R}$ holds within the ball $B(\Theta_0, R)$ for some $R > 0$ and $\mu > 0$, where $L_g$ is the Lipschitz constant of $g$, then the tangent kernel $K(\Theta)$ is $\mu$-uniformly conditioned on the ball $B(\Theta_0, R)$. Hence, the square loss satisfies the $\mu$-PL* condition in $B(\Theta_0, R)$.*

**Lemma A.5** (Theorem 3.2 of Liu et al. (2020)). *Consider a neural network $g_\Theta$ of the form Eq. (1). Let $w$ be the minimum of the hidden layer widths, i.e., $w = \min_{j\in l} p^j$. Given any fixed $R > 0$, and any $\Theta \in B(\Theta_0, R)$, with high probability over the initialization, the Hessian spectral norm satisfies the following:*

$$\|H_g(\Theta)\| = \tilde{O}(R^{3l}/\sqrt{w}).$$

**Lemma A.6** (Singular value of product of matrices). *Suppose two matrices $P$ and $Q$ satisfies $P \in \mathbb{R}^{m\times n}$, $Q \in \mathbb{R}^{n\times s}$, let $\lambda_i(\cdot)$ and $\lambda_{\min}(\cdot)$ be the i-th and minimal singular value of one matrix, respectively, then*

$$\lambda_i(PQ) \geq \lambda_{\min}(P)\lambda_i(Q).$$

*Proof.*

$$\lambda_i(PQ) = \max_{S:\dim(S)=i} \min_{\substack{x\in S,\\\|x\|=1}} \|PQx\|$$

$$\geq \lambda_{\min}(P) \cdot \max_{S:\dim(S)=i} \min_{\substack{x\in S,\\\|x\|=1}} \|Qx\|$$

$$= \lambda_{\min}(P)\lambda_i(Q).$$

□

Then we start our proof. Before projection, GaLore has the same tangent kernel $K(\Theta)$ and Hessian matrix $H_g$ as full-rank method. According to Lemma A.5, under the given condition, $\|H_g(\Theta)\| = \tilde{O}(R^{3l}/\sqrt{w})$. Directly plugging this into the condition of Lemma A.4, let $\epsilon = \lambda_0 - \mu$, we get $K(\Theta)$ is $\mu$-uniformly conditioned on the ball $B(\Theta_0, R)$.

During the projection process, GaLore first project the gradient matrix to a low-rank space, and then projects it back to full-rank space. Therefore, if $\widehat{K(\Theta)}$ is still $\mu$-uniformly conditioned on the ball $B(\Theta_0, R)$, where $\widehat{K(\Theta)}$ is the tangent kernel after the projection process, the conclusion of Theorem 4.2 holds for GaLore based on the fact $K(\Theta)$ is $\mu$-uniformly conditioned. In the follows, we analyze the singular values of $\widehat{K(\Theta)}$. The analysis is divided into two steps, the first step is to only consider one-layer neural network, and the second step is to generalize the result to multi-layer neural networks.

**Step 1**: For one-layer neural network, suppose the gradient matrix for the i-th sample $G_i \in \mathbb{R}^{p \times q}$, where $p < q$. GaLore performs SVD to $G_0$.

$$G_0 = U \Sigma V^T,$$

where $U \in \mathbb{R}^{p \times p}$ and $V \in \mathbb{R}^{q \times q}$ are orthogonal matrices, $\Sigma \in \mathbb{R}^{p \times q}$ is a diagonal matrix. For rank r, the projection matrix is selected as $\hat{U} = U[: r] \in \mathbb{R}^{r \times p}$, i.e., the first three rows of original $U$. Then $G$ is projected by $\hat{U}$ and projected back by $\hat{U}^T$. After the projection process, the gradient matrix $G_i$ becomes

$$\hat{G}_i = \hat{U}^T \hat{U} G_i.$$

According to the definition of $K(\Theta)$, $K(\Theta) = D_{g\Theta}(D_{g\Theta})^T$, and

$$D_{g\Theta} = \begin{bmatrix} \text{vec}(G_0) \\ \text{vec}(G_1) \\ \vdots \\ \text{vec}(G_{m-1}) \end{bmatrix},$$

where $\text{vec}(G_i)$ is the row vector transformed from matrix $G_i$. Thus, $\text{vec}(G_i) = \mathbb{R}^{pq}$ and $D_{g\Theta} = \mathbb{R}^{m \times pq}$. After the projection process, $\widehat{D_{g\Theta}}$ becomes

$$\widehat{D_{g\Theta}} = D_{g\Theta} = \begin{bmatrix} \text{vec}(\hat{G}_0) \\ \text{vec}(\hat{G}_1) \\ \vdots \\ \text{vec}(\hat{G}_{m-1}) \end{bmatrix}.$$

Let $I_q$ be the $q \times q$ identity matrix and $\otimes$ be the Kronecker product. Then we have

$$\widehat{D_{g\Theta}} = D_{g\Theta} \left( I_q \otimes \hat{U} \right)^T.$$

According to the definition of Kronecker product, $I_q \otimes \hat{U}$ results in a matrix consisting of $q$ copies of the matrix $\hat{U}$ arranged along the diagonal, with all other positions filled with zeros. Therefore, $I_q \otimes \hat{U} \in \mathbb{R}^{rq \times pq}$.

Note that $\hat{U}$ is the first r rows of orthogonal matrix $U$, $\lambda_i(\hat{U}) = 1$ for $i \leq r$. Therefore, $\lambda_i(I_q \otimes \hat{U}) = 1$ for $i \leq rq$. According to Lemma A.6,

$$\begin{aligned} \lambda_{\min}(\widehat{D_{g\Theta}}) &= \lambda_n(\widehat{D_{g\Theta}}) \\ &= \lambda_n(D_{g\Theta} \left( I_q \otimes \hat{U} \right)^T) \\ &\geq \lambda_{\min}(D_{g\Theta})\lambda_n \left( \left( I_q \otimes \hat{U} \right)^T \right) \\ &\geq \mu. \end{aligned} \tag{4}$$

The last inequality comes from $\lambda_{\min}(D_{g_\Theta}) = \lambda_{\min}(K(\Theta)) \geq \mu$ and the assumption $n \leq rq$. Thus, $\widehat{K(\Theta)}$ is also $\mu$-uniformly conditioned and then conclude the proof.

**Step 2**: For multi-layer neural network, the gradient matrix is not only one $G_i$ but a series of matrices $G_i^0, G_i^1, \ldots, G_i^l$, where $l$ is the number of layers. Let $\hat{U}_j$ be the projection matrix for $G_i^j$, in this situation,

$$\widehat{D_{g_\Theta}} = D_{g_\Theta} \begin{bmatrix} (I_q \otimes \hat{U}_0)^T & 0 & \cdots & 0 \\ 0 & (I_q \otimes \hat{U}_1)^T & \cdots & 0 \\ \vdots & \vdots & \ddots & \vdots \\ 0 & 0 & \cdots & (I_q \otimes \hat{U}_l)^T \end{bmatrix}.$$

For such a block diagonal matrix, the blocks are independent of each other, allowing for analysis of each block individually. According to the conclusion of Step 1, $\lambda_n(\widehat{D_{g_\Theta}}) \geq \mu$ if $n \leq r(p^l + p^{l-1} + \cdots + p^1)$.

### A.1.3 PROOF OF THEOREM 4.4

When applied LoRA, the network defined by Eq. (1) becomes

$$g_\Theta[\mathbf{x}] := \Theta_2^l I[\Theta_1^l f^l[\Theta_2^{l-1} I[\Theta_1^{l-1} \ldots f^1[\Theta_2^0 I[\Theta_1^0 \mathbf{x}]]], \tag{5}$$

where $\Theta_2^j \in \mathbb{R}^{p^j \times r}$, $\Theta_1^j \in \mathbb{R}^{r \times p^{j-1}}$ and $I$ is identity function. Apply Lemma A.5 to Eq. (5), we have

$$\|H_g(\Theta)\| = \tilde{O}(R^{6l}/\sqrt{r}).$$

Combine this result with Lemma A.4, we get the final result.

## A.2 PROOF OF THEOREM 4.9

We follow the analysis framework of Lederer (2020). In this section, we introduce essential concepts related to path relations and block parameters.

**Definition A.7** (Path relations). Consider two parameters $\Theta, \Gamma \in \mathcal{M}$. If there is a continuous function $h_{\Theta,\Gamma} : [0,1] \to \mathcal{M}$ that satisfies $h_{\Theta,\Gamma}[0] = \Theta$, $h_{\Theta,\Gamma}[1] = \Gamma$, and $t \mapsto \ell(g_{h_{\Theta,\Gamma}[t]})$ is constant, we say that $\Theta$ and $\Gamma$ are path constant and write $\Theta \leftrightarrow \Gamma$.

If there is a continuous function $h_{\Theta,\Gamma} : [0,1] \to \mathcal{M}$ that satisfies $h_{\Theta,\Gamma}[0] = \Theta$, $h_{\Theta,\Gamma}[1] = \Gamma$, and $t \mapsto \ell[g_{h_{\Theta,\Gamma}[t]}]$ is convex, we say that $\Theta$ and $\Gamma$ are path convex and write $\Theta \rightsquigarrow \Gamma$.

If there are parameters $\Theta', \Gamma' \in \mathcal{M}$ such that (i) $\Theta \leftrightarrow \Theta'$ and $\Gamma \leftrightarrow \Gamma'$ and (ii) $\Theta' \rightsquigarrow \Gamma'$, we say that $\Theta$ and $\Gamma$ are path equivalent and write $\Theta \leftrightsquigarrow \Gamma$.

**Definition A.8** (Block parameters). Consider a number $s \in \{0, 1, \ldots\}$ and a parameter $\Theta \in \mathcal{M}$. If

1. $(\Theta^j)_i = 0$ for all $j > s$;

2. $(\Theta^v)_{ji} = 0$ for all $v \in \{1, \ldots, l-1\}$ and $i > s$ and for all $v \in \{1, \ldots, l-1\}$ and $j > s$;

3. $(\Theta^l)_{ij} = 0$ for all $j > s$,

we call $\Theta$ an $s$-upper-block parameter of depth $l$.

Similarly, if

1. $(\Theta^j)_i = 0$ for all $j \leq p^1 - s$;

2. $(\Theta^v)_{ji} = 0$ for all $v \in \{1, \ldots, l-1\}$ and $i \leq p^{v+1} - s$ and for all $v \in \{1, \ldots, l-1\}$ and $j \leq p^v - s$;

3. $(\Theta^l)_{ij} = 0$ for all $j \leq p^l - s$,

we call $\Theta$ an $s$-lower-block parameter of depth $l$. We denote the sets of the $s$-upper-block and $s$-lower-block parameters of depth $l$ by $\mathcal{U}_{s,l}$ and $\mathcal{L}_{s,l}$, respectively. According to the definition of block parameters, $\mathcal{U}_{\mu,l} \in \mathcal{M}_\mu$. Then we propose the following proposition.

**Proposition A.9.** *For every $\Theta \in \mathcal{M}_\mu$, there are $\overline{\Theta}, \underline{\Theta} \in \mathcal{M}$ with $\overline{\Theta} \in \mathcal{U}_{\mu,l}$ and $\underline{\Theta} \in \mathcal{L}_{\mu,l}$ such that $\Theta \leftrightarrow \overline{\Theta}$ and $\Theta \leftrightarrow \underline{\Theta}$.*

The proposition implies every parameter $\Theta \in \mathcal{M}_\mu$, it is path constant to a $\mu$-upper-block parameter and a $\mu$-lower-block parameter. To prove this proposition, we introduce two lemmas. The first lemma is proposed by Lederer (2020), while the second lemma will be proven here.

**Lemma A.10** (Lemma 3 of Lederer (2020)). *Consider permutations $p^j : \{1, \ldots, p^j\} \to \{1, \ldots, p^j\}$ for $j \in \{0, \ldots, l+1\}$. Assume that $p^0$ and $p^{l+1}$ are the identity functions: $p^0[j] = p^{l+1}[j] = j$ for all $j$. The parameter $\Theta \in \mathcal{M}$ is a spurious local minimum of the objective function (2) if and only if $\Gamma \in \mathcal{M}$ defined through $(\Gamma^j)_{uv} := (\Theta^j)_{p^j[v]}$ for all $j \in \{0, \ldots, l\}, u \in \{1, \ldots, p^j\}$ and $v \in \{1, \ldots, p^j\}$ is a spurious local minimum of the objective function (2).*

**Lemma A.11.** *Consider three matrices $A \in \mathbb{R}^{u \times v}$, $B \in \mathbb{R}^{v \times r}$, and $C \in \mathbb{R}^{r \times o}$, and a function $h : \mathbb{R} \to \mathbb{R}$. With some abuse of notation, define $h : \mathbb{R}^{v \times r} \to \mathbb{R}^{v \times r}$ through $(h[M])_{ij} := h[M_{ij}]$ for all $M \in \mathbb{R}^{v \times r}$. Assume the rank of $BC$ is no larger than $\mu$, then, there are matrices $A \in \mathbb{R}^{u \times v}$ and $B \in \mathbb{R}^{v \times o}$ and a permutation $p : \{1, \ldots, v\} \to \{1, \ldots, v\}$ such that*

- $\overline{A}[h\overline{B}[C]] = A_p[h[B^p[C]]]$;

- $\overline{A}_{ij} = 0$ *for* $j > \mu$; $B_{ji} = 0$ *for* $j > \mu$ *and* $B_{ji} = (B^p)_{ji}$ *otherwise.*

*Similarly, there are matrices $A \in \mathbb{R}^{u \times v}$ and $B \in \mathbb{R}^{v \times o}$ and a permutation $p : \{1, \ldots, v\} \to \{1, \ldots, v\}$ such that*

- $\underline{A}[h\underline{B}[C]] = A_p[h[B^p[C]]]$;

- $\underline{A}_{ij} = 0$ *for* $j \leq v - \mu$; $B_{ji} = 0$ *for* $j \leq v - \mu$ *and* $B_{ji} = (B^p)_{ji}$ *otherwise.*

*Proof.* **Step 1:** Fix a $k \in \{1, \ldots, u\}$. We first show that there is a matrix $\dot{A} \in \mathbb{R}^{u \times v}$ such that

1. $\dot{A}h[BC] = Ah[BC]$;

2. $\#\{j \in \{1, \ldots, v\} : \dot{A}_{kj} \neq 0\} \leq \mu$;

3. $\dot{A}_{aj} = A_{aj}$ for all $a \neq k$.

For every $k \in \{1, \ldots, u\}$ and $i \in \{1, \ldots, r\}$, elementary matrix algebra yields that

$$(Ah[BC])_{ki} = \sum_{j=1}^{v} A_{kj}(h[BC])_{ji}.$$

Denoting the row vectors of a matrix $M \in \mathbb{R}^{a \times b}$ by $M_{1\bullet}, \ldots, M_{a\bullet} \in \mathbb{R}^b$, we then get

$$(Ah[BC])_{k\bullet} = \sum_{j=1}^{v} A_{kj}(h[BC])_{j\bullet}.$$

That is, $(Ah[BC])_{k\bullet}$ lies in the space spanned by $\{(h[BC])_{j\bullet}\}_{j=1}^{v}$. Because the rank of $BC$ is no more than $m$, there is at most $\mu$ $(h[BC])_{j\bullet}$ are linearly independent. Suppose the linearly independent vectors are $\{(h[BC])_{j_s\bullet}\}_{s=1}^{\mu}$, $(Ah[BC])_{k\bullet}$ can be rewritten as:

$$(Ah[BC])_{k\bullet} = \sum_{s=1}^{\mu} t_{j_s}(h[BC])_{j_s\bullet}.$$

Define

$$\dot{A}_{aj} := \begin{cases} t_{j_s} & \text{for } a = k, j = j_s; \\ 0 & \text{for } a = k, j \neq j_s; \\ A_{aj} & \text{otherwise.} \end{cases}$$

Then $\dot{A}_{aj}$ follows the above three properties.

**Step 2:** We then show that there is a matrix $\widetilde{A} \in \mathbb{R}^{u \times v}$ such that

1. $\widetilde{A}h[BC] = Ah[BC]$;

2. $\#\{j \in \{1, \ldots, v\} : \widetilde{A}_{kj} \neq 0, \forall k \in \{1, \ldots, u\}\} \leq \mu$.

Since Step 1 changes only the kth row of A, we can apply it to one row after another.

**Step 3:** By Property 3 of the previous step, the matrix $\widetilde{A}$ has at most $\mu$ nonzero columns. Verify that replacing $A$ by $A_p$ and $B$ by $B_p$ for a suitable permutation $p$ leads to $\widetilde{A}$ whose entries outside the first $\mu$ columns are equal to zero. We denote this version of $\widetilde{A}$ by $\overline{A}$. Note that

$$
\begin{aligned}
(h[B^p C])_{ji} = h\left([B^p C]_{ji}\right) &= h\left(\left[\sum_{b=1}^{o}(B^p)_{jb}C_{bi}\right]\right) \\
&= h\left(\left[\sum_{b=1}^{o}(C^T)_{ib}(B^p)_{jb}\right]\right) = h\left([(C^T(B^p))_{j\bullet}]_i\right) \\
&= (h[C^T(B^p)])_{ji},
\end{aligned}
$$

Combining this result with the results of Step 2 (with $A$ and $B$ replaced by $A_p$ and $B^p$) yields for all $a \in \{1, \ldots, u\}$ and $i \in \{1, \ldots, r\}$ that

$$
\begin{aligned}
(A_p h[B^p C])_{ai} = \sum_{j=1}^{v}(A_p)_{aj}(h[B^p C])_{ji} &= \sum_{j=1}^{v}(A_p)_{aj}(h[C^T(B^p)_{j\bullet}])_i \\
&= \sum_{j=1}^{v}\overline{A}_{aj}(h[C^T(B^p)_{j\bullet}])_i = \sum_{j=1}^{\min\{\mu,v\}}\overline{A}_{aj}(h[C^T(B^p)_{j\bullet}])_i.
\end{aligned}
$$

We then define $\overline{B} \in \mathbb{R}^{v \times o}$ through

$$
\overline{B}_{ji} := \begin{cases} (B^p)_{ji} = B_{p[j]i} & \text{for } j \leq \mu; \\ 0 & \text{otherwise.} \end{cases}
$$

Then we have

$$
\begin{aligned}
(A_p h[B^p C])_{ai} &= \sum_{j=1}^{\min\{\mu,v\}}\overline{A}_{aj}(h[C^T(B^p)_{j\bullet}])_i \\
&= \sum_{j=1}^{\min\{\mu,v\}}\overline{A}_{aj}(h[C^T B_{j\bullet}])_i = \sum_{j=1}^{\min\{\mu,v\}}\overline{A}_{aj}(h[BC])_{ji} \\
&= \sum_{j=1}^{v}\overline{A}_{aj}(h[BC])_{ji} = (\overline{A}h[\overline{B}C])_{ai}.
\end{aligned}
$$

The second part of the lemma can be derived in the same way.

$\square$

Then we start to prove Proposition A.9.

*Proof of Proposition A.9.* To facilitate the proof, we define the data matrix $X \in \mathbb{R}^{d \times n}$ through $X_{ji} := (x_i)_j$ for all $j \in \{1, \ldots, d\}$ and $i \in \{1, \ldots, n\}$, that is, each column of $X$ consists of one sample. We finally write

$$g_\Theta[X] := (g_\Theta[x_1], \ldots, g_\Theta[x_n]) = \Theta^l f^l[\Theta^{l-1} \ldots f^1[\Theta^0 X]] \in \mathbb{R}^{m \times n}$$

for all $\Theta \in \mathcal{M}$. Hence, $g_\Theta[X]$ summarizes the network's outputs for the given data.

Given a parameter $\Theta \in \mathcal{M}_\mu$, we establish a corresponding upper-block parameter $\overline{\Theta} \in \mathcal{U}_{s,l}$ layer by layer, starting from the outermost layer. We write

$$g_\Theta[X] = \underbrace{\Theta^l}_{=:A \in \mathbb{R}^{p^{l+1} \times p^l}} \underbrace{f^l}_{=:h} [\underbrace{\Theta^{l-1}}_{=:B \in \mathbb{R}^{p^l \times p^{l-1}}} \underbrace{f^{l-1}[\Theta^{l-2} \ldots f^1[\Theta^0 X]]]}_{=:C \in \mathbb{R}^{p^{l-1} \times n}}.$$

Lemma A.11 for two-layer networks then gives

$$g_\Theta[X] = \overline{\Theta}^l f^l \left[ \begin{pmatrix} \overline{\Theta}^{l-1} \\ 0 \end{pmatrix} \right] f^{l-1}[\Theta^{l-2} \ldots f^1[\Theta^0 X]].$$

for a matrix $\overline{\Theta}^l \in \mathbb{R}^{p^{l+1} \times p^l}$ that meets Condition 3 in the first part of Def. A.8 on block parameters as long as $s \geq \mu$ and for a matrix $\overline{\Theta}^{l-1} \in \mathbb{R}^{\mu \times p^{l-1}}$ that consists of the first $\mu$ rows of the matrix $\Theta^{l-1}$. Here we ignore the permutation in Lemma A.11 because of the symmetric property identified by Lemma A.10. (We implicitly assume here and in the following $p^j \geq \mu$ for all $j \in \{1, \ldots, l\}$.)

Now, define a parameter $\Gamma^l \in \mathcal{M}_\mu$ through

$$\Gamma^l := (\overline{\Theta}^l, \Theta^{l-1}, \ldots, \Theta^0)$$

and a function $h_{\Theta, \Gamma^l} : [0, 1] \to \mathcal{M}_\mu$ through

$$h_{\Theta, \Gamma^l}[t] := (1 - t)\Theta + t\Gamma^l \text{ for all } t \in [0, 1].$$

The function $h_{\Theta, \Gamma^l}$ is continuous and satisfies $h_{\Theta, \Gamma^l}[0] = \Theta$ and $h_{\Theta, \Gamma^l}[1] = \Gamma^l$. Moreover, we can 1. use the definitions of the function $h_{\Theta, \Gamma^l}$ and the networks, 2. split the network along the outermost layer, 3. invoke the block shape of $\overline{\Theta}^l$ and the definition of $\overline{\Theta}^{l-1}$ as the $\mu$ first rows of the matrix $\Theta^{l-1}$, 4. use the above-stated inequalities for the network $g_\Theta[X]$, and 5. consolidate the terms to show for all $t \in [0, 1]$ that

$$g_{\Theta, \Gamma^l}[t][X] = ((1 - t)\Theta^l + t\overline{\Theta}^l) f^l[\Theta^{l-1} \ldots f^1[\Theta^0 X]]$$
$$= (1 - t)\Theta^l f^l[\Theta^{l-1} \ldots f^1[\Theta^0 X]] + t\overline{\Theta}^l f^l[\Theta^{l-1} \ldots f^1[\Theta^0 X]]$$
$$= (1 - t)\Theta^l f^l[\Theta^{l-1} \ldots f^1[\Theta^0 X]] + t\overline{\Theta}^l f^l \left[ \begin{pmatrix} \overline{\Theta}^{l-1} \\ 0 \end{pmatrix} \ldots f^1[\Theta^0 X] \right]$$
$$= (1 - t)g_\Theta[X] + tg_\Theta[X]$$
$$= g_\Theta[X].$$

Hence, the function $t \mapsto \ell(g_{h_{\Theta, \Gamma^l}[t]})$ is constant. Suppose $\Theta^l = U\Sigma$. That is, $\Theta$ and $\Gamma^l$ are path constant: $\Theta \leftrightarrow \Gamma^l$.

We then move one layer inward. Similar to the above derivation, we have

$$\overline{\Theta}^{l-1} f^{l-1}[\Theta^{l-2} \ldots f^1[\Theta^0 X]] = \overline{\Theta}^{l-1} f^{l-1} \left[ \begin{pmatrix} \overline{\Theta}^{l-2} \\ 0 \end{pmatrix} \ldots f^1[\Theta^0 X] \right],$$

for a matrix $\overline{\Theta}^{l-1} \in \mathbb{R}^{\mu \times p^{l-1}}$ that meets Condition 3 in the first part of Def. A.8 on block parameters as long as $s \geq \mu$, and for a matrix $\overline{\Theta}^{l-2} \in \mathbb{R}^{\mu \times p^{l-2}}$ that consists of the first $\mu$ rows of the matrix $\Theta^{l-2}$.

Next, we define $\overline{\Theta}^{l-1} \in \mathbb{R}^{p^l \times p^{l-1}}$ through

$$(\overline{\Theta}^{l-1})_{uv} := \begin{cases} (\overline{\Theta}^{l-1})_{uv} & \text{for } u \leq \mu \\ 0 & \text{otherwise.} \end{cases}$$

Combining this definition with the above-derived results yields

$$g_\Theta[X] = \Theta^l f^l [\overline{\Theta}^{l-1} f^{l-1} \left[ \begin{pmatrix} \overline{\Theta}^{l-2} \\ 0 \end{pmatrix} \ldots f^1 [\Theta^0 X] \right] ],$$

and the matrix $\overline{\Theta}^{l-1}$ meets Condition 2 in the first part of Def. A.8 on block parameters as long as $s \geq \mu$.

Similarly as above, define a parameter $\Gamma^{l-1} \in \mathcal{M}$ through

$$\Gamma^{l-1} := (\overline{\Theta}^l, \overline{\Theta}^{l-1}, \Theta^{l-2}, \ldots, \Theta^0)$$

and a function $h_{\Gamma^l, \Gamma^{l-1}} : [0, 1] \to \mathcal{M}$ through

$$h_{\Gamma^l, \Gamma^{l-1}}[t] := (1 - t)\Gamma^l + t\Gamma^{l-1} \text{ for all } t \in [0, 1]$$

to show that $\Gamma^l \leftrightarrow \Gamma^{l-1}$. In view of Property 3 in Lemma 1, we can conclude that $\Theta \leftrightarrow \Gamma^{l-1}$.

The proof can be finished by induction over the layers.

$\square$

We state two additional propositions by Lederer (2020), they are necessary to complete our proof.

**Proposition A.12** (Proposition 3 of Lederer (2020)). *Consider two block parameters $\Theta \in \mathcal{U}_{s,l}$ and $\Gamma \in \mathcal{L}_{s,l}$. If $w \geq 2s$, it holds that $\Theta \leftrightsquigarrow \Gamma$.*

**Proposition A.13** (Prop 1 of Lederer (2020)). *Assume that for all $\Theta \in \mathcal{M}$, there is a global minimum of the objective function (4), denoted by $\Gamma$, such that $\Theta \leftrightsquigarrow \Gamma$. Then, the objective function (4) has no spurious local minima.*

Let the global minima is $\Gamma$. For any parameters $\Theta \in \mathcal{M}_\mu$, according to Proposition A.9, there exists $\overline{\Theta} \in \mathcal{U}_{\mu,l}$ such that $\Theta \leftrightarrow \overline{\Theta}$. Similarly, there exists $\underline{\Gamma} \in \mathcal{L}_{\mu,l}$, such that $\Gamma \leftrightarrow \underline{\Gamma}$.

According to Proposition A.12, $\overline{\Theta} \leftrightsquigarrow \underline{\Gamma}$. Therefore, $\Theta \leftrightsquigarrow \Gamma$. Note that this relationship holds for all $\Theta \in \mathcal{M}_\mu$, we conclude that the objective function (2) has no projected spurious local minima. Then we complete the proof of Theorem 4.9.

## A.3 PROOF OF SECTION 5

### A.3.1 PROOF OF LEMMA 5.1

To prove that a random matrix $R \in \mathbb{R}^{m \times r}$ (with $r < m$) whose entries are independently drawn from a Gaussian distribution $\mathcal{N}(0, \sigma^2)$ has full rank $r$ with probability 1, we need to show that the probability of $\text{rank}(R) < r$ is zero. Here's how we can establish this:

A matrix $R$ has rank less than $r$ if and only if there exists a non-zero vector $x \in \mathbb{R}^r$ such that:

$$Rx = 0$$

This equation represents a homogeneous linear system. For $R$ to have a non-trivial null space, the columns of $R$ must be linearly dependent.

The set of rank-deficient matrices can be characterized by the vanishing of all $r \times r$ minors (determinants of $r \times r$ submatrices). These minors are polynomial functions of the entries of $R$:

$$\det(R_k) = 0 \quad \text{for all} \quad k$$

where $R_k$ denotes the $r \times r$ submatrix formed by selecting any $r$ rows from $R$.

The set where these determinants vanish is an algebraic variety defined by polynomial equations. In $\mathbb{R}^{m \times r}$, such a set has Lebesgue measure zero.

Since the set of rank-deficient matrices has Lebesgue measure zero and the distribution of $R$ is absolutely continuous (Gaussian distribution), the probability that $R$ is rank-deficient is zero:

$$P(\text{rank}(R) < r) = 0$$

Therefore, the probability that $R$ has full rank is:

$$P(\text{rank}(R) = r) = 1 - P(\text{rank}(R) < r) = 1$$

### A.3.2 PROOF OF COROLLARY 5.2

First, we need to introduce the following lemma.

**Lemma A.14** (Marchenko-Pastur Law). *Suppose a matrix $X \in \mathbb{R}^{n \times p}$, where the entries $X_{ij}$ are independent and identically distributed with mean 0 and variance $\sigma^2$. As $n$ and $p$ tend to infinity, with the ratio $n/p$ converging to a constant $\lambda$, the empirical spectral distribution (i.e., the distribution of eigenvalues) of the matrix $\frac{1}{p} X X^T$ converges to a limit given by the Marchenko-Pastur distribution. This distribution has a density function given by:*

$$f(x) = \frac{1}{2\pi\sigma^2 x}\sqrt{(b-x)(x-a)} \quad \text{for } a \leq x \leq b$$

*where*
$$a = \sigma^2(1 - \sqrt{\lambda})^2, \quad b = \sigma^2(1 + \sqrt{\lambda})^2.$$

The difference between GaRare and GaLore is the selection of projection matrix. GaRare uses a random matrix $R$ instead of $\hat{U}$. Thus, Eq. (4) becomes

$$\lambda_m(\widehat{D_{g\Theta}}) = \lambda_m(D_{g\Theta}(I_n \otimes R)^T) \geq \lambda_{\min}(D_{g\Theta})\lambda_m\left((I_n \otimes R)^T\right).$$

Similar to the proof of Theorem 4.3, $\lambda_{\min}(D_{g\Theta}) \geq \mu$. Then we need to determine the singular value of $I_n \otimes R$.

For $R \in \mathbb{R}^{m \times r}$, we can apply Lemma A.14 by replacing $n$ and $p$ with $m$ and $r$, $\sigma^2$ with $1/m$, the eigenvalues of $\frac{1}{r} R R^T$ converges to $f(x) = \frac{m}{2\pi x}\sqrt{(b-x)(x-a)}$ for $a \leq x \leq b$, with $a = \frac{1}{m}(1 - \sqrt{\lambda})^2$, $b = \frac{1}{m}(1 + \sqrt{\lambda})^2$.

Because we are considering large language models, the actual distribution is similar to the given one. Therefore, $\lambda_m(R) \geq ra = \frac{1}{\lambda}(1 - \sqrt{\lambda})^2$ with high probability. Note that $r \ll m$, then $ra \approx 1$. Then we have $\lambda_m(R) \geq 1$ with high probability.

Remember that $I_n \otimes R$ is a matrix consisting of $n$ copies of the matrix $R$ arranged along the diagonal, with all other positions filled with zeros, we have $\lambda_m(I_n \otimes R) \geq 1$ with high probability. Then with high probability,

$$\lambda_m(\widehat{D_{g\Theta}}) \geq \mu.$$

The remaining proofs are the same as the proof of Theorem 4.3.

### A.3.3 PROOF OF COROLLARY 5.3

The proof of Theoem 4.9 only uses the fact that the solution of GaLore is located in low-rank spaces. Thus, use $R$ instead of $\hat{U}$ does not change the conclusion.

Table 5: Architectures of the MLPs in Section 6.1.

| Parameter | Value |
|---|---|
| input_dim | $32 \times 32 \times 3$ |
| hidden_dim1 | hidden_dim |
| hidden_dim2 | hidden_dim |
| output_dim | 10 |

| Parameter | Value |
|---|---|
| input_dim | $32 \times 32 \times 3$ |
| hidden_dim1 | hidden_dim |
| hidden_dim2 | rank |
| hidden_dim3 | hidden_dim |
| output_dim | 10 |

# B  DETAILS OF EXPERIMENTS

## B.1  MODEL ARCHITECTURE OF VERIFICATION EXPERIMENT

The architecture of the MLPs in Section 6.1 is in Table B.1. On the left is the architecture of full-rank method, FLoRA, GaLore and GaRare, while on the right is the architecture of LoRA. Because the weight matrices of LoRA is frozen, we omit them and only include trainable parameters in the architecture. *hidden_dim* and *rank* changed across different tasks according to the experiment setting.

## B.2  ARCHITECTURE AND HYPER-PARAMETERS OF PRE-TRAINING EXPERIMENT

We introduce details of the LLaMA architecture and hyperparameters used for pre-training. Table 6 shows the most hyperparameters of LLaMA models across model sizes. We follow the setting of (Zhao et al., 2024), which uses a max sequence length of 256 for all models, with a batch size of 131K tokens. For all experiments, we adopt learning rate warmup for the first 10% of the training steps, and use cosine annealing for the learning rate schedule, decaying to 10% of the initial learning rate.

Table 6: Model architecture comparison across different parameter sizes.

| Params | Hidden | Intermediate | Heads | Layers | Steps | Data amount |
|---|---|---|---|---|---|---|
| 60M | 512 | 1376 | 8 | 8 | 10K | 1.3 B |
| 130M | 768 | 2048 | 12 | 12 | 20K | 2.6 B |
| 350M | 1024 | 2736 | 16 | 24 | 60K | 7.8 B |
| 1 B | 2048 | 5461 | 24 | 32 | 100K | 13.1 B |
| 7 B | 4096 | 11008 | 32 | 32 | 30K | 3.9 B |

For GaRare and FLoRA on each size of models (from 60M to 1B), we tune their favorite learning rate from a set of {0.04, 0.05, 0.03, 0.02, 0.01}, and the best learning rate is chosen based on the validation perplexity. For the other methods, full-rank method, LoRA and GaLore, we use results reported by Zhao et al. (2024) directly. The hyperparameters of GaRare is shown in Table 7.

Table 7: Hyperparameters of GaRare across different parameter sizes.

| Params | Learning rate | LoRA scale | Subspace update frequency |
|---|---|---|---|
| 60M | 0.04 | 0.1 | 500 |
| 130M | 0.03 | 0.1 | 500 |
| 350M | 0.03 | 0.1 | 500 |
| 1 B | 0.02 | 0.1 | 500 |
| 7 B | 0.005 | 0.01 | 500 |

## B.3 DETAILS OF FINE-TUNING ON GLUE

We fine-tune the pre-trained RoBERTa-Base and RoBERTa-Large model on the GLUE benchmark using the model provided by the Hugging Face[1]. We tune the learning rate and batch size for GaRare. Table 8 and 9 show the hyperparameters used for fine-tuning RoBERTa-Base and RoBERTa-Large for GaRare, respectively.

Table 8: Hyperparameter settings for each dataset for RoBERTa-Base model.

|  | MNLI | SST-2 | MRPC | CoLA | QNLI | QQP | RTE | STS-B |
|---|---|---|---|---|---|---|---|---|
| Batch Size | 16 | 16 | 16 | 32 | 8 | 8 | 16 | 16 |
| # Epochs | 30 | 30 | 30 | 30 | 30 | 30 | 30 | 30 |
| Learning Rate | 1E-05 | 2E-05 | 3E-05 | 3E-04 | 1E-05 | 3E-05 | 4E-05 | 3E-04 |
| Rank Config. | | | | $r = 4$ | | | | |
| scale $\alpha$ | | | | 4 | | | | |
| Max Seq. Len. | | | | 512 | | | | |

Table 9: Hyperparameter settings for each dataset for RoBERTa-Large model.

|  | MNLI | SST-2 | MRPC | CoLA | QNLI | QQP | RTE | STS-B |
|---|---|---|---|---|---|---|---|---|
| Batch Size | 8 | 16 | 8 | 32 | 16 | 8 | 16 | 16 |
| # Epochs | 30 | 30 | 30 | 30 | 30 | 30 | 30 | 30 |
| Learning Rate | 1E-05 | 3E-05 | 1E-04 | 3E-05 | 1E-05 | 1E-05 | 1E-04 | 3E-05 |
| Rank Config. | | | | $r = 8$ | | | | |
| scale $\alpha$ | | | | 4 | | | | |
| Max Seq. Len. | | | | 512 | | | | |

## C PSEUDO-CODE OF GARARE

---

**Algorithm 1: GaRare**

**Input:** $\Theta \in \mathcal{M}$, rank $r \in \mathbb{N}^+$, scale factor $\alpha$, subspace update frequency $T$, random seed $x$, MaxIter > 0.

**Initialize:** First-order moment $M_0 = 0$; Second-order moment $V_0 = 0$

1 **for** $k = 1$ **to** *MaxIter* **do**
2      **if** *k% T = 0* **then**
3          x+=1
4      Compute loss function according to Eq. (2) with parameters $\Theta$ and get gradients $G_t \in \mathbb{R}^{m \times n}$;
5      Generate a random matrix $R \in \mathbb{R}^{m \times r}$ sampled from $\mathcal{N}(0, 1/m)$;
6      $\hat{G}_t = R^T g_t$;
7      Get parameters update $\widehat{\Delta g_t}$ by optimization method (e.g., Adam, Adafactor);
8      $\Delta G_t = R \widehat{\Delta G_t}$;
9      $\Theta = \Theta + \Delta G_t$;

**Output:** $W$

---

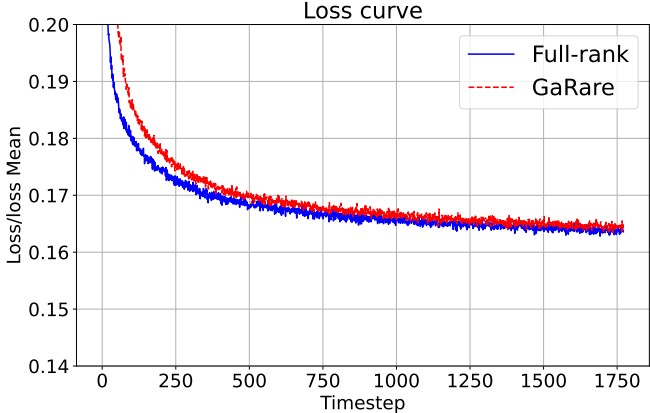

Figure 3: The training curve of EDM on full-rank training and GaRare.

## D  COMPUTATIONAL COST

Table 10: Compuational time of full-rank training, GaRare, GaLore, FLoRA and LoRA on various sizes of LLaMA models on C4 dataset.

|  | 60M (1gpu) | 130M (2gpu) | 350M (2gpu) | 1B (4gpu) |
|---|---|---|---|---|
| Full-Rank | 1.98h | 6.30h | 26.47h | 133.54h |
| LoRA | 1.93h | 5.73h | 24.87h | 127.14h |
| FLoRA | 1.97h | 6.00h | 26.44h | 129.88h |
| GaLore | 1.92h | 6.07h | 26.98h | 131.55h |
| GaRare | 1.97h | 5.96h | 26.50h | 130.00h |
| $r/d_{\text{model}}$ | 128 / 256 | 256 / 768 | 256 / 1024 | 512 / 2048 |
| Training Tokens | 1.1B | 2.2B | 6.4B | 13.1B |

## E  GARARE IN EDM

To evaluate the generalizability of GaRare, we applied it to optimize EDM (Karras et al., 2022), a widely used diffusion model. The model was trained on the CIFAR10-32x32 dataset, with the loss curve presented in Figure 3. While GaRare shows a relatively slower convergence rate, its final performance is comparable to that of full-rank training. Notably, GaRare reduces memory usage to 564M compared to 1024M for full-rank training.

## F  SENSITIVITY ANALYSIS

We analyzed the sensitivity of GaRare to random seeds and subspace update frequency. First, we used three random seeds to run the experiments and reported the mean and standard deviation, as shown in Table G and G, to demonstrate the robustness of GaRare with respect to random seeds. Second, we evaluated the sensitivity of GaRare to the subspace update frequency. Following the approach of GaLore Zhao et al. (2024), we conducted experiments on LLaMA 130M using the C4 dataset, with the results presented in Figure 4. The performance remains stable when the update frequency exceeds 100, indicating that this hyperparameter can be selected with a relatively large value without significantly impacting performance.

# G   MEMORY USAGE ANALYSIS

In this section, we analyze the memory usage of LoRA, GaLore, and GaRare by discussing weights and optimizer states separately.

**Weights**

- **LoRA**: Requires storing the low-rank matrices $B \in \mathbb{R}^{p \times r}$ and $A \in \mathbb{R}^{r \times q}$, along with the full-rank weight matrix $\Theta \in \mathbb{R}^{p \times q}$. This results in a total of $pq + pr + qr$ weights.
- **GaLore and GaRare**: Both eliminate the need to store the low-rank matrices $B$ and $A$, requiring only $pq$ weights for the forward computation.

**Optimizer States**

- **LoRA**: Requires storing gradient momentum and variance for $B$ and $A$, adding $2pr + 2qr$ to the total memory usage.
- **GaLore**: Stores optimizer states in the low-rank spaces and for the projection matrix, resulting in a total of $pr + 2qr$.
- **GaRare**: Reduces memory usage for optimizer states to $2qr$ by regenerating the projection matrix at each iteration, avoiding the need for additional storage.

This breakdown highlights how GaRare matches GaLore in weights but achieves superior memory efficiency in optimizer states.

Table 11: The mean and standard deviation of GaRare on various sizes of LLaMA models on C4 dataset.

|  | 60M | 130M | 350M | 1B |
|---|---|---|---|---|
| Full-Rank | 34.06 | 25.08 | 18.80 | 15.56 |
| LoRA | 34.99 | 33.92 | 25.58 | 19.21 |
| FLoRA | 36.97 | 30.22 | 22.67 | 20.22 |
| GaLore | 34.88 | **25.36** | **18.95** | **15.64** |
| GaRare | **34.33±0.09** | 25.49±0.20 | 19.24±0.07 | 15.69±0.14 |
| $r/d_{\mathrm{model}}$ | 128 / 256 | 256 / 768 | 256 / 1024 | 512 / 2048 |
| Training Tokens | 1.1B | 2.2B | 6.4B | 13.1B |

Table 12: The mean and standard deviation of GaRare with RoBERTa-Base and RoBERTa-Large model on GLUE.

|  | RoBERTa-Base | | | | | RoBERTa-Large | | | | |
|---|---|---|---|---|---|---|---|---|---|---|
|  | Full-Rank | GaRare | GaLore | FLoRA | LoRA | Full-Rank | GaRare | GaLore | FLoRA | LoRA |
| Memory | 748M | 252M | 253M | 252M | 257M | 2132M | 718M | 720M | 718M | 732M |
| **CoLA** | 62.2 | 61.0±0.1 | 60.4 | 59.0 | **61.4** | 68.0 | 67.6±0.4 | **68.3** | 65.5 | 68.2 |
| **STS-B** | 90.9 | 90.3±0.0 | **90.7** | 89.9 | 90.6 | 91.5 | 92.3±0.0 | 92.5 | 92.5 | **92.6** |
| **MRPC** | 91.3 | 91.4±0.2 | **92.3** | 88.5 | 91.1 | 90.9 | **91.7±0.1** | 91.7 | 89.3 | 90.9 |
| **RTE** | 79.4 | **79.2±0.3** | 79.4 | 76.5 | 78.7 | 86.6 | **87.2±0.3** | 87.0 | 83.0 | 87.4 |
| **SST2** | 94.6 | **94.2±0.2** | 94.0 | 93.8 | 92.9 | 96.4 | **96.0±0.3** | 96.1 | 96.0 | 96.2 |
| **MNLI** | 87.2 | **87.0±0.2** | 87.0 | 86.6 | 86.8 | 90.2 | **91.2±0.4** | 90.8 | 90.4 | 90.6 |
| **QNLI** | 92.3 | **92.1±0.2** | 92.2 | 91.9 | 92.2 | 94.7 | 94.5±0.1 | **95.7** | 94.0 | 94.9 |
| **QQP** | 92.3 | 90.9±0.3 | 91.1 | 90.9 | **91.3** | 92.2 | **91.9±0.2** | 91.9 | 91.5 | 91.5 |
| **Avg** | 86.3 | 85.8 | **85.9** | 84.6 | 85.6 | 88.8 | 89.1 | **89.3** | 87.8 | 89.0 |

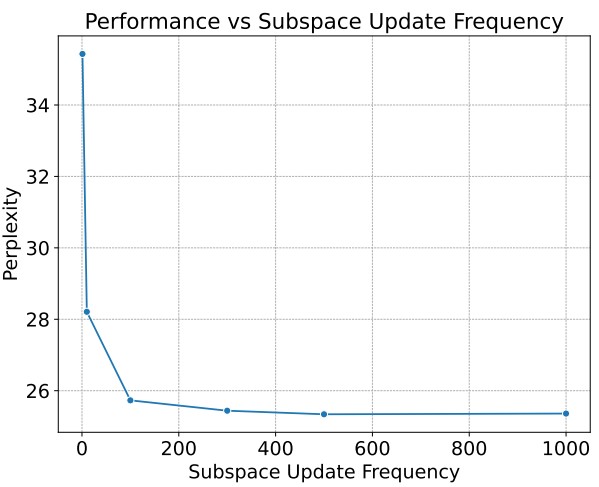

Figure 4: Ablation study of GaLore on 130M models on subspace update frequency.

