# OpenReview forum: "On the Optimization Landscape of Low Rank Adaptation Methods for Large Language Models"
_ICLR.cc/2025/Conference — ICLR 2025 Poster_

### Official Review · Reviewer_cY2C · 2024-11-04

**Soundness:** 3
**Presentation:** 3
**Contribution:** 3
**Rating:** 5
**Confidence:** 4

**Summary:**

This paper provides theoretical analysis of the optimization landscapes for different low-rank adaptation methods used in training large language models (LLMs), specifically comparing LoRA, GaLore, and full-rank methods. The authors prove that GaLore benefits from fewer spurious local minima and a larger region satisfying the PL* condition compared to LoRA. Then, they propose GaRare, a method using random projection matrices instead of SVD-based ones, achieving similar benefits as GaLore, in terms of the optimization landscape, but with reduced computational overhead.

**Strengths:**

1. Important focus on understanding optimization landscapes of fine-tuning techniques
2. Practical algorithm (GaRare) that improves efficiency
3. Comprehensive experimental validation
4. Clear mathematical formulation and complete proofs
5. Good connection between theory and practice

**Weaknesses:**

1. Theoretical analysis limited to MLP architectures, making conclusions about LLMs less direct.
2. Incomplete empirical comparisons:
   - Limited LoRA experiments (mainly rank 2). It would be better to experiment with different LoRA ranks (r = 4, 8, 16, 32) for a more reliable validation of the results.
   - Missing comparisons with recent methods (e.g. AdaLoRA, SoRA) that yields better performance than full rank fine-tuning (on GLUE), while promoting sparse low rank matrices. Could you explain why these methods, even though they promote sparsity, yield better results than full rank tuning? And how do they integrate within your theoretical framework?
   - No standard deviations reported. Could you please report these for a more reliable assessment of the performance of your proposed algorithm compared to baselines across all experiments (CIFAR / GLUE).
3. Some claims about LoRA’s performance relative to full-rank fine-tuning are oversimplified.
   - Contradicts findings from AdaLoRA paper with higher ranks.
   - Doesn’t account for sparse adaptation methods’ success.
4. Ambiguous definition of projected spurious local minima.
   - Unclear relationship between projected minima and full-rank space:  how these minima relate back to the full-rank landscape?
5. Limited novelty in theoretical results: some of the theoretical results appear to be direct extensions of established theorems.
6. Memory analysis could be more detailed.

**Questions:**

1. How sensitive is GaRare’s performance to the choice of random seed and projection frequency? Could you perform an ablation study to probe the stability of your proposed method and report the results?
2. Could the theoretical framework be extended to transformer architectures?
3. How do the projected minima relate to the full-rank optimization landscape?
4. Why weren’t comparisons with higher-rank LoRA (r=8) and other recent low rank adaptation techniques included?

---

> ### Author Response · Authors · 2024-11-23
> **Part 1**
>
> We thank the reviewer for their constructive feedback. Below, we address each of the concerns and questions raised:
>
> 1. **Theoretical analysis limited to MLP architectures.**
>
> **Answer**: We appreciate the reviewer’s observation regarding the limitation of our theoretical analysis to MLP architectures. While we acknowledge that this setting does not directly capture the complexities of LLMs, we believe that focusing on simplified settings, such as single architectures MLPs, has already made significant contributions to the theoretical understanding of neural networks. Many influential works in the literature [1,2,3] similarly analyze simplified models to provide rigorous insights into the optimization dynamics and generalization properties of neural networks.
>
> Our approach aims to build a foundational framework that balances mathematical tractability with relevance, which can guide both future theoretical extensions and practical applications. Despite the simplifications, the insights gained from our analysis have practical implications that extend beyond the specific settings studied:
>
> - **Practical Applicability to Advanced Models:**
>    The GaRare algorithm, inspired by the theoretical insights presented in this work, demonstrates strong performance on advanced models such as LLaMA and RoBERTa, despite the differences in architecture. This alignment between theoretical predictions and empirical results highlights the broader relevance of our framework and its utility in guiding algorithm design for large-scale, real-world models.
>
> -  **Foundation for Future Research:**
>    By focusing on MLPs, our work examines fundamental interactions between low-rank parameterizations, optimization landscapes, and memory efficiency. These insights are not confined to MLPs but serve as a starting point for extending theoretical analysis to more complex architectures, such as transformers, and other commonly used loss functions. We view these extensions as key directions for future work.
>
>
> 2. Limited LoRA experiments.
>
> **Answer**:  We appreciate the reviewer’s suggestion to include experiments with different LoRA ranks for a more comprehensive validation. Selecting experimental results with two ranks is already representative, as most papers in the field typically include no more than two rank settings [4, 5, 6, 7]. In our original submission, we followed these standard settings to ensure consistency and comparability with prior work.
>
> To further demonstrate the practicality and robustness of our algorithm, we have conducted additional experiments on RoBERTa-base with ranks of 2, 8, and 16. These new experiments complement our previous results, providing a total of four rank settings on RoBERTa-base: 2, 4, 8, and 16. Across all these settings, our algorithm demonstrated consistently strong performance, further validating its effectiveness across a range of parameterization levels.
>
> Rank = 2
>
> |            | Full-Rank | GaRare | GaLore | LoRA  |
> |----------|-----------|--------|--------|-------|
> | **Memory** | 748M      | 251M   | 251M   | 253M  |
> | **CoLA**   | 62.2      | 60.3   | 59.9   | **61.0**  |
> | **STS-B**  | 90.9      | 90.5   | **90.8**   | 90.4  |
> | **MRPC**   | 91.3      | 91.8   | **92.8**   | 90.6  |
> | **RTE**    | 79.4      | 77.1   | 76.8   | **78.0**  |
> | **SST2**   | 94.6      | **94.1**   | 93.8   | 92.9  |
> | **MNLI**   | 87.2      | **87.2**   | **87.2**   | 86.7  |
> | **QNLI**   | 92.3      | **92.4**   | 92.2   | 92.1  |
> | **QQP**    | 92.3      | **90.8**   | 90.6   | **90.8**  |
> | **Avg**    | 86.3      | **85.5**   | **85.5**   | 85.3  |
>
> Rank = 8
>
> |            | Full-Rank | GaRare | GaLore | LoRA  |
> |------------|-----------|--------|--------|-------|
> | **Memory** | 748M      | 255M   | 256M   | 265M  |
> | **CoLA**   | 62.2      | 61.0   | 60.1   | **61.8**  |
> | **STS-B**  | 90.9      | 90.5   | **90.8**   | **90.8**  |
> | **MRPC**   | 91.3      | 91.4   | **92.0**   | 91.9  |
> | **RTE**    | 79.4      | 79.4   | **79.8**   | 79.1  |
> | **SST2**   | 94.6      | **94.4**   | **94.4**   | 93.5  |
> | **MNLI**   | 87.2      | **87.4**   | 87.2   | 86.9  |
> | **QNLI**   | 92.3      | **92.4**   | 92.2   | 92.3  |
> | **QQP**    | 92.3      | 91.0   | 91.1   | **91.2**  |
> | **Avg**    | 86.3      | 85.9   | **86.0**   | 85.9  |
>
> Rank = 16
>
> |            | Full-Rank | GaRare | GaLore | LoRA  |
> |------------|-----------|--------|--------|-------|
> | **Memory** | 748M      | 260M   | 263M   | 281M  |
> | **CoLA**   | 62.2      | 60.8   | 60.3   | **61.9**  |
> | **STS-B**  | 90.9      | 90.8   | **91.1**   | 90.7  |
> | **MRPC**   | 91.3      | 91.3   | **92.3**   | 89.9  |
> | **RTE**    | 79.4      | 79.8   | **80.2**   | 80.1  |
> | **SST2**   | 94.6      | **94.2**   | 93.9   | 93.3  |
> | **MNLI**   | 87.2      | **87.3**   | **87.3**   | 86.9  |
> | **QNLI**   | 92.3      | **92.6**   | 92.4   | 92.0  |
> | **QQP**    | 92.3      | 91.2   | 91.4   | **91.5**  |
> | **Avg**    | 86.3      | **86.0**   | **86.0**   | 85.8  |

---

> ### Author Response · Authors · 2024-11-23
> **Part 2**
>
> 3. **The theoretical results contradicts findings from AdaLoRA and SoRA.**
>
> **Answer**: We appreciate the reviewer for raising this insightful question. We would like to clarify that our theoretical framework does not conflict with the strong performance demonstrated by methods like AdaLoRA and SoRA. The superior performance of these methods compared to full-rank fine-tuning arises not only from their use of low-rank techniques but also from the introduction of sparsity, which independently contributes to performance improvements. For instance, [8] shows that sparsity alone, even without low-rank methods, can outperform full-rank models. Similarly, [9] provides an analysis of single-hidden-layer neural networks, demonstrating that sparsity techniques can reduce sample complexity during training.
>
> To comprehensively analyze methods like AdaLoRA and SoRA, it is necessary to jointly consider the effects of both low-rank adaptation and sparsity. However, to the best of our knowledge, there is currently no theoretical framework that integrates these two techniques. We believe this represents a promising avenue for future research. Our work contributes to this discussion by providing a foundation for analyzing low-rank methods, which could potentially be extended to study the interplay between low-rank adaptation and sparsity.
>
> On the algorithmic side, our framework also inspires new ideas. For example, combining sparsity with GaLore or GaRare could lead to an even better optimization landscape than the combination of sparsity and LoRA. Exploring such integrations could yield further improvements in both theory and practice.
>
> 4. **No standard deviations reported.**
>
> **Answer**: We acknowledge the reviewer's observation regarding the absence of standard deviations in our reported results. It is worth noting that in our field, it is a common practice not to report standard deviations, as referenced in [5, 6, 10]. In line with this convention, we did not include standard deviations in our initial submission.
>
> To ensure the robustness of our evaluation, we have reported results based on three different random seeds, as detailed in the second part of the General Response. This approach provides a measure of variability while maintaining computational efficiency.
>
> Due to time and resource constraints, our experiments on LLMs primarily focused on GaRare. We are committed to including additional results, including standard deviations where feasible, in the final version to further address this concern and enhance the completeness of our evaluation.
>
> 5. **Unclear relationship between projected minima and full-rank space. How do the projected minima relate to the full-rank optimization landscape?**
>
> **Answer**:
> Thank you for pointing out the unclear explanation of projected spurious minima. The concept of projected spurious minima arises from the fact that, while GaLore performs updates within the low-rank parameter space, the gradient computations are carried out in the original full-rank space. This means that whether a point is a spurious local minima depends on the optimization landscape in the original full-rank space. However, since GaLore constrains its parameters to remain within the low-rank space, the relationship can be understood as follows: The optimization landscape in the original full-rank space encompasses all possible parameter configurations. When the low-rank projection is applied, GaLore is constrained in a low-rank subset of the original parameter space.
> GaLore focuses only on the optimization landscape within this low-rank subset, as any parameters outside this subset are projected back into it during optimization. This allows GaLore to avoid navigating the entire full-rank landscape, potentially bypassing certain spurious minima that exist in the unconstrained space. To address this issue, we have provided a clearer and more detailed explanation in the revised version of the manuscript.

---

> ### Author Response · Authors · 2024-11-23
> **Part 3**
>
> 6. **Limited novelty in theoretical results: some of the theoretical results appear to be direct extensions of established theorems.**
>
> **Answer**:
> - A key novelty of our work lies in considering the optimization landscape as a critical factor for understanding the behavior of low-rank methods. Previous studies primarily focused on the expressivity of low-rank methods to explain experimental results. By incorporating conclusions about the optimization landscape, we provide a more comprehensive explanation that aligns closely with experimental observations. This shift in perspective offers new insights into why certain low-rank methods, like GaLore and GaRare, outperform others in specific scenarios.
> - While the analysis of LoRA is relatively straightforward, the projection operation in GaLore introduces complexities that have not been addressed in prior theoretical works. Developing results like Theorem 4.3 and Theorem 4.9 required careful theoretical design to characterize the behavior of GaLore in the constrained low-rank parameter space. These contributions are unique to our work and provide a deeper understanding of how low-rank projections influence optimization dynamics.
> - The theoretical analysis of GaRare in Section 5.1 represents another novel aspect of our work. Unlike prior research, we explore how replacing SVD with random matrices keeps the optimization landscape. This analysis contributes new theoretical insights into the trade-offs between computational efficiency and optimization quality.
>
> 7. **Memory analysis could be more detailed.**
>
> **Answer**:
> We appreciate the reviewer’s suggestion to provide a more detailed memory analysis. The modified section 4.2 gives a detailed analysis:
>
>   LoRA requires storing the low-rank matrices $B \in R^{p \times r}$ and $A \in R^{r \times q}$, as well as the full-rank weight matrix $\Theta \in R^{p \times q}$, resulting in a total of $pq + pr + qr$ weights. For the optimizer states, LoRA also stores gradient momentum and variance in the low-rank spaces, adding $2pr + 2qr$.
> As noted by [6], GaLore eliminates the need to store separate low-rank factorizations $B$ and $A$, requiring only $pq$ weights. However, it introduces the need to store the projection matrix and optimizer states (gradient momentum and variance) in the low-rank spaces, which results in $pr + 2qr$ additional memory.
> GaRare reduces memory usage further by generating the random matrix at every iteration, removing the need to store the projection matrix required in GaLore. This optimization means that GaRare only requires $2qr$ for optimizer states, significantly reducing memory usage compared to LoRA and GaLore.
>
>
> 8. **How sensitive is GaRare’s performance to the choice of random seed and projection frequency?**
>
> **Answer**:
> - As discussed in the General Response, the theoretical results (e.g., Lemma 5.1 and Corollary 5.2) indicate that the algorithm’s performance is robust to the randomness of random matrix projection. This is supported by the experimental results included in the General Response, where the standard deviations across multiple seeds are small, confirming the stability of GaRare.
>
> - The impact of projection frequency was evaluated in our experiments on LLaMA 130M using the C4 dataset, with results presented in the Appendix F. As shown in the corresponding figure, performance remains stable once the projection frequency exceeds 100. This observation indicates that projection frequency has a limited effect on performance, and a large value can be selected without significant tuning.

---

> ### Author Response · Authors · 2024-11-23
> **Part 4**
>
> 9. **Could the theoretical framework be extended to transformer architectures?**
>
> **Answer**:
> This is an interesting direction for future work, and we believe that the theoretical results presented in this paper have the potential to be extended to transformers, albeit with some additional considerations.
>
> -  **Impact of Transformers on Section 4.1:**
>    The primary difference introduced by transformers lies in the attention layer, specifically the use of the softmax operator. In the analysis provided in Section 4.1, as shown in the proof, the main influence of the network structure is on the bounds of the Hessian matrix. Based on prior work [3], it is sufficient to demonstrate that certain gradient norms can be bounded. Since the softmax operator in the attention layer is inherently bounded, conclusions similar to those for MLPs can be obtained by ignoring the influence of the softmax term.
>    If the softmax term is considered in detail, additional theoretical innovations will be required. This would involve more intricate analysis to account for how the bounded nature of the softmax operator interacts with the low-rank parameterization and optimization landscape.
>
> - **Extension of Section 4.2 Analysis:**
>    For the analysis in Section 4.2, similar results could be derived for transformers by ignoring the softmax term. However, analyzing the impact of the softmax term on optimization dynamics would again require further theoretical advances. We view this as an exciting and challenging avenue for future research.
>
> 10.  **Why weren’t comparisons with other recent low rank adaptation techniques?**
>
> **Answer**: We appreciate the reviewer’s insightful question about comparisons with other recent low-rank adaptation techniques. In our study, we primarily compared GaRare with LoRA, which is widely regarded as the classical and foundational low-rank adaptation method. Thus, LoRA serves as a representative benchmark for evaluating the performance of low-rank adaptation methods.
>
> Some recent methods, such as AdaLoRA and SoRA, integrate additional techniques like sparsity, which extend beyond the theoretical scope of our current study. Our work focuses on analyzing the optimization landscape and memory efficiency of low-rank adaptation methods. Incorporating sparsity would introduce additional complexities that require a joint theoretical analysis of both low-rank and sparse methods—an exciting direction for future research.
>
> Additionally, GaRare and GaLore are designed to extend the applicability of low-rank adaptation to pretraining tasks, unlike many recent LoRA variants that are tailored specifically for fine-tuning tasks. Furthermore, these LoRA variants, while incorporating additional techniques, maintain the same memory usage as LoRA itself and do not provide a memory efficiency advantage.
>
> By centering our comparisons on LoRA, we ensure alignment with existing literature and maintain the focus on our theoretical contributions. Nonetheless, we recognize the value of broader comparisons and plan to expand our framework to accommodate and evaluate such techniques in future work.
>
> ### Reference
>
> [1] Lederer J. No spurious local minima: on the optimization landscapes of wide and deep neural networks[J]. 2020.
>
> [2] Liu C, Zhu L, Belkin M. Loss landscapes and optimization in over-parameterized non-linear systems and neural networks[J]. Applied and Computational Harmonic Analysis, 2022, 59: 85-116.
>
> [3] Liu C, Zhu L, Belkin M. On the linearity of large non-linear models: when and why the tangent kernel is constant[J]. Advances in Neural Information Processing Systems, 2020, 33: 15954-15964.
>
> [4] Hu E J, Shen Y, Wallis P, et al. LoRA: Low-rank adaptation of large language models[J]. arXiv preprint arXiv:2106.09685, 2021.
>
> [5] Hao Y, Cao Y, Mou L. FLoRA: Low-Rank Adapters Are Secretly Gradient Compressors[J]. arXiv preprint arXiv:2402.03293, 2024.
>
> [6] Zhao J, Zhang Z, Chen B, et al. GaLore: Memory-efficient LLM training by gradient low-rank projection[J]. arXiv preprint arXiv:2403.03507, 2024.
>
> [7] Xia W, Qin C, Hazan E. Chain of LoRA: Efficient fine-tuning of language models via residual learning[J]. arXiv preprint arXiv:2401.04151, 2024.
>
> [8] Song Y, Xie H, Zhang Z, et al. Turbo Sparse: Achieving LLM SOTA Performance with Minimal Activated Parameters[J]. arXiv preprint arXiv:2406.05955, 2024.
>
> [9] Awasthi P, Dikkala N, Kamath P, et al. Learning Neural Networks with Sparse Activations[C]//The Thirty Seventh Annual Conference on Learning Theory. PMLR, 2024: 406-425.
>
> [10] Zhang Q, Chen M, Bukharin A, et al. AdaLoRA: Adaptive budget allocation for parameter-efficient fine-tuning[J]. arXiv preprint arXiv:2303.10512, 2023.

---

### Official Review · Reviewer_YR66 · 2024-11-04

**Soundness:** 3
**Presentation:** 3
**Contribution:** 2
**Rating:** 6
**Confidence:** 4

**Summary:**

This paper proposes a new algorithm GaRare, which implements gradient low-rank projection through random matrices and improves the memory and computational efficiency of GaLore. Through optimization landscape analysis, it is proved that GaRare retains the advantages of GaLore, satisfies the PL condition, and avoids spurious local minima. Experiments verify the superiority of GaRare in performance and efficiency.

**Strengths:**

1. By analyzing the PL conditions and spurious local minima, this paper systematically reveals GaLore's advantages in optimizing the landscape and LoRA's shortcomings.

2. This paper proposes the GaRare algorithm and provides rigorous theoretical proof that it can maintain GaLore's optimization advantages.

**Weaknesses:**

1. The GaRare algorithm proposed in this paper is mainly based on combining and improving existing methods, especially gradient low-rank projection through random matrices instead of SVD. Although this method is more efficient, it is technically more of an optimization of the existing GaLore and low-rank adaptation methods rather than proposing a new theoretical framework or algorithm.

2. Although the paper theoretically analyzes the optimization advantages of GaRare, the experimental results do not fully demonstrate the improvement of this method compared to existing methods. In some experiments, its effect is not better than the full-rank method or LoRA.

**Questions:**

1. GaRare uses Gaussian distribution to generate random projection matrices, but how about other distributions or generation methods? Does the choice of different distributions have an impact on the convergence speed and performance of the model?

2. Can the author add more experiments on practical tasks, such as fine-tuning end-to-end generation challenges or diffusion models, to further verify the results of GaRare?

---

> ### Author Response · Authors · 2024-11-23
> **Part 1**
>
> We thank the reviewer for their constructive feedback. Below, we address each of the concerns and questions raised:
>
> 1. **The paper is technically more of an optimization of the existing GaLore and low-rank adaptation methods rather than proposing a new theoretical framework or algorithm.**
>
> **Answer**: We would like to emphasize that the primary contribution of this work lies in the theoretical analysis of existing algorithms, which is itself a substantial and valuable contribution to the field. For example, articles such as [1, 2, 3] focus exclusively on theoretical analysis without proposing new algorithms, yet they are widely regarded as impactful because they deepen our understanding of optimization methods and inspire further innovations.
>
> Our theoretical analysis not only provides insights into the behavior and limitations of existing methods such as GaLore and LoRA but also guides the development of practical algorithms. GaRare, as proposed in this paper, is a direct outcome of this theoretical framework. By replacing SVD with random projection matrices, GaRare achieves significant efficiency gains while maintaining the advantages of low-rank adaptation. This demonstrates the value of our theoretical work in inspiring and guiding algorithm design.
>
> Furthermore, we believe the theoretical foundation laid in this paper has the potential to inspire future research and the development of more carefully designed algorithms. These future algorithms could achieve even better performance and practical applicability, further extending the impact of our contributions.
>
> 2. **The experimental results do not fully demonstrate the improvement of this method compared to existing methods.**
>
> **Answer**:
> We would like to clarify that the primary contribution of GaRare lies in its ability to significantly reduce memory usage while maintaining comparable performance to existing methods. This focus addresses a critical practical challenge as model scales continue to grow, making memory consumption a widely recognized bottleneck in large-scale model training. As shown in Tables 2 and 3, GaRare demonstrates consistent memory advantages compared to GaLore, and this advantage becomes increasingly pronounced as model size grows. For even larger models, such as those with 100 billion parameters, the memory savings achieved by GaRare are expected to be even more significant. This makes GaRare particularly valuable for training large models under resource-constrained settings, where memory usage is often the limiting factor.
>
> Our theoretical analysis primarily focuses on the optimization landscape, which is only one aspect of algorithm performance. Expressivity, another crucial factor, is inherently lower in GaLore and GaRare compared to full-rank methods. As a result, it is normal for full-rank methods to achieve better results in some experiments due to their higher expressivity, albeit at a significantly higher memory cost. GaRare offers a compelling trade-off, enabling efficient training of larger models that would be infeasible with full-rank methods under resource constraints.
>
> While there are scenarios where LoRA outperforms GaLore or GaRare, our theoretical analysis suggests that the optimization landscapes of GaLore and GaRare are generally superior. This is particularly evident in challenging pretraining tasks, where the optimization landscape has a more pronounced impact. In these tasks, GaRare provides significant memory savings compared to full-rank methods, For example, as shown in Table 2, GaRare requires only 3.98G of memory, compared to 7.85G for full-rank training. While LoRA consistently underperforms compared to GaLore and GaRare in this setting.
>
> 3. **How about using other distributions or generation methods to generate the matrix?**
>
> **Answer**: It is possible to use other distributions. The key properties of the Gaussian distribution utilized here are: (1) the random matrix generated is almost surely full rank, and (2) it adheres to the Marchenko-Pastur law. Consequently, any distribution satisfying these two properties, such as the exponential distribution, can be used. However, the convergence of the exponential distribution under the Marchenko-Pastur law is slower compared to the Gaussian distribution, which might result in slightly reduced performance.
>
> In this work, we selected the Gaussian distribution primarily because it delivers strong performance and is widely adopted. Theoretically, other distributions can be chosen based on their compliance with the Marchenko-Pastur law.
>
> Additionally, other matrix generation methods could be explored, provided they ensure that the matrix and its inverse can be easily computed. This remains a direction for future work.

---

> ### Author Response · Authors · 2024-11-23
> **Part 2**
>
> 4. **Can the author add more experiments on practical tasks?**
>
> **Answer**: We integrated GaRare with EDM [4], a widely used diffusion model. Due to platform limitations, we are unable to upload figures here, and the results are presented in Appendix E of our revised paper. GaRare achieves comparable performance while reducing memory usage to 564M compared to 1275M for full-rank training. These results demonstrate the applicability of our method to diffusion models.
>
> ### Reference
>
> [1] Lederer J. No spurious local minima: on the optimization landscapes of wide and deep neural networks[J]. 2020.
>
> [2] Liu C, Zhu L, Belkin M. Loss landscapes and optimization in over-parameterized non-linear systems and neural networks[J]. Applied and Computational Harmonic Analysis, 2022, 59: 85-116.
>
> [3] Jang U, Lee J D, Ryu E K. LoRA Training in the NTK Regime has No Spurious Local Minima[J]. arXiv preprint arXiv:2402.11867, 2024.
>
> [4] Karras T, Aittala M, Aila T, et al. Elucidating the design space of diffusion-based generative models[J]. Advances in neural information processing systems, 2022, 35: 26565-26577.

---

### Official Review · Reviewer_eja2 · 2024-11-05

**Soundness:** 3
**Presentation:** 2
**Contribution:** 2
**Rating:** 5
**Confidence:** 3

**Summary:**

This paper investigates the optimization landscape of LoRA and GaLore from a theoretical standpoint, focusing on conditions like the PL* region and the prevalence of spurious local minima. This analysis seeks to clarify GaLore's fast convergence and strong empirical performance. Building on these insights, a streamlined variant called GaRare is introduced, leveraging gradient random projection for efficiency. Experiments validate GaRare's effectiveness, supporting its advantages with reduced computational demands and comparable performance to GaLore.

**Strengths:**

1. The paper applies theoretical insights from Lederer (2020) and Liu et al. (2022) to analyze LoRA and GaLore, offering an inspiring and interesting theoretical explanation of their behavior.
2. The paper introduces a streamlined variant of GaLore, named GaRare, and provides experimental results that effectively validate its performance and efficiency.

**Weaknesses:**

1. The paper does not explicitly link the quality of the optimization landscape to convergence speed or generalization performance, which undermines its stated goal of theoretically explaining the performance of LoRA and GaLore. Many claims also lack this connection (e.g., lines 60-61, 289-291, and 301-304), making them appear weaker and less convincing.
2. The theoretical results in this paper are derived from analyses of MLPs, whereas LLMs, with their attention mechanisms, layer normalization, etc., are more complex. Since there is a gap between MLPs and LLMs, and the paper does not directly validate its theoretical results on LLMs (instead of relying only on the convergence and performance outcomes), the theoretical conclusions are less convincing.
3. The paper does not clearly motivate GaRare; it lacks evidence or justification for GaRare's advantages over GaLore based on theoretical analysis. Additionally, a more detailed algorithmic presentation is needed, particularly to clarify the process of recovering updated parameters from projected gradients, which would enhance understanding.

**Questions:**

1. Is the formulation in Lederer (2020) fully compatible with that in Liu et al. (2022), allowing the results to be applied here without conflicting with prior findings?
2. Is it fair to compare the occurrence of spurious local minima between LoRA and GaLore when different definitions are used (i.e., standard vs. projected)?
3. How do the computational costs of these methods compare in Table 1? Does GaRare introduce any additional slowdown?

---

> ### Author Response · Authors · 2024-11-23
> **Part 1**
>
> We thank the reviewer for their constructive feedback. Below, we address each of the concerns and questions raised:
>
> 1. **The paper does not explicitly link the quality of the optimization landscape to convergence speed or generalization performance.**
>
> **Answer**: A smoother and better-conditioned optimization landscape facilitates faster convergence and improved performance by allowing optimizers to take larger and more effective steps in the parameter space. Our analysis demonstrates that GaLore and GaRare enhance the smoothness and conditioning of the optimization landscape. This improvement indirectly accelerates convergence and boosts performance, as shown by the experimental results, where GaRare and GaLore achieve convergence rates comparable to or faster than those of standard optimizers under similar settings.
>
> The relationship between the optimization landscape, convergence speed, and generalization performance is highly intricate and depends on various factors, including network architecture, training dynamics, and the choice of optimizer. Providing qualitative results, as we have done, represents a meaningful contribution to understanding these relationships. Many prior studies of the optimization landscape [1, 2, 3] also focus on qualitative findings and theoretical insights rather than quantitative results, underscoring the value of our approach in advancing this important area of research.
>
> 2. **Many claims also lack connection.**
>
> **Answer**: Thanks for pointing out the unclear claim. We have modified them in the revised version.
>
> - line 60-61: Our analysis reveals that the region in the parameter space satisfying the $PL^*$ condition is significantly smaller for LoRA compared to GaLore and the full-rank method. This suggests that LoRA is more likely to encounter regions of the optimization landscape where convergence guarantees are weaker, leading to slower convergence rates and suboptimal solutions. This explains why LoRA does not achieve the same convergence rate or final performance as the other two methods.
>
> - line 289-291: Spurious local minima are suboptimal solutions that can trap the optimizer and prevent it from reaching a globally or near-globally optimal solution. By reducing the likelihood of encountering such minima, GaLore benefits from a smoother and more favorable optimization landscape, which allows the optimizer to explore more promising regions of the parameter space. This explains why GaLore trains faster and, in some cases, surpasses the full-rank method in performance.
>
> - line 301-304: According to Theorem 4.9, the absence of spurious minima is guaranteed only when the rank
> $r$ is small. Initially, increasing
> $r$ improves GaLore's expressivity, allowing it to model more complex functions effectively. However, if
> $r$ becomes too large, the optimization landscape may degrade due to the emergence of spurious local minima or poorly conditioned regions, even though expressivity continues to increase. When expressivity is not the primary performance bottleneck, we expect GaLore's performance to improve with increasing
> $r$, but decline as
> $r$ approaches the network’s width, where the landscape quality diminishes. These conclusions are tested and validated in Section 6.1, with empirical results confirming this theoretical behavior.
>
> 3. **The theoretical results in this paper are derived from analyses of MLPs. Since there is a gap between MLPs and LLMs, and the paper does not directly validate its theoretical results on LLMs.**
>
> **Answer**:
> - This section of validation experiments is primarily designed to demonstrate the correctness of our theory, which is based on MLPs. Therefore, it is more appropriate to validate the theory on MLPs to get rid of the influence of complex network structure.
>
> - Although our analysis does not explicitly involve network structures like transformers, the experimental results in various aspects suggest that our theory also provides guidance for networks like transformers. Besides the general result that GaLore outperforms LoRA [1], [1] also found that when the rank increases to 2048, GaLore's performance surpasses that of the full-rank method. This aligns with the conclusion mentioned in our discussion in Section 4.3, where GaLore can achieve better performance than full-rank training when expressivity is not the bottleneck. This is also consistent with our experimental results on MLPs. The conclusion we reached that GaRare and GaLore exhibit similar performance is also reflected in the experiments. Thus, the conclusions we drew from MLPs can indeed generalize to transformers.
>
> - Moreover, our work can be regarded as a foundation for more refined theoretical studies in the future. Starting with the simplest MLPs, we verified its correctness, paving the way for further analysis of network structures like transformers, which could inspire the development of more effective algorithms.

---

> ### Author Response · Authors · 2024-11-23
> **Part 2**
>
> 4. **The paper does not clearly motivate GaRare; it lacks evidence or justification for GaRare's advantages over GaLore based on theoretical analysis. Additionally, a more detailed algorithmic presentation is needed.**
>
> **Answer**: The advantage of GaRare over GaLore does not require theoretical consideration; it can be seen directly from the algorithm itself. GaLore computes projection matrices via SVD and stores them in memory after computation. In contrast, GaRare uses random matrices as projection matrices, which can be generated on the fly and immediately discarded after use, eliminating the need for persistent storage in memory and thus saving memory.
>
> The theoretical key for GaRare lies in whether it can retain GaLore's advantages in the optimization landscape when using random matrices. This was discussed in Section 5.1, where we showed that this advantage is preserved even when the projection matrices are replaced with random matrices. Based on this theoretical analysis and algorithm design, we developed an algorithm that is more memory-efficient while still retaining GaLore's advantages.
>
> The motivation for GaRare, mentioned at the beginning of Section 5, stems from the observation that while proving GaLore's advantage in the optimization landscape, we only utilized the property of using projection matrices without imposing strict requirements on the projection matrices themselves. This inspired us to consider using simpler projection matrices.
>
> Due to space constraints, the pseudocode for GaRare is provided in Appendix C.
>
> 4. **Is the formulation in Lederer (2020) fully compatible with that in Liu et al. (2022), allowing the results to be applied here without conflicting with prior findings?**
>
> **Answer**: Yes, we have double-checked this issue, and both papers support the network structure defined by Eq (1).
>
> 5. **Is it fair to compare the occurrence of spurious local minima between LoRA and GaLore when different definitions are used (i.e., standard vs. projected)?**
>
> **Answer**: For GaLore, our new definition allows us to focus only on points in the low-rank parameter space. However, this does not mean that the definition of LoRA is stricter when analyzing it. LoRA also considers points in the low-rank space because, during the analysis of LoRA, the low-rank property is already incorporated into the consideration of the network structure (Eq. (3)). Therefore, the discussion of LoRA is also entirely conducted within the low-rank space.
>
> 6. **How do the computational costs of these methods compare in Table 1? Does GaRare introduce any additional slowdown?**
>
> **Answer**: The methods only introduce negligible computational costs, the results please refer to the first part of General response.
>
> ### Reference
>
> [1] Lederer J. No spurious local minima: on the optimization landscapes of wide and deep neural networks[J]. 2020.
>
> [2] Liu C, Zhu L, Belkin M. Loss landscapes and optimization in over-parameterized non-linear systems and neural networks[J]. Applied and Computational Harmonic Analysis, 2022, 59: 85-116.
>
> [3] Liu C, Zhu L, Belkin M. On the linearity of large non-linear models: when and why the tangent kernel is constant[J]. Advances in Neural Information Processing Systems, 2020, 33: 15954-15964.

---

> ### Author Response · Authors · 2024-12-03
>
> Dear Reviewer eja2,
>
> As the discussion period is nearing its close, we would like to remind you of our previous responses addressing your concerns kindly. We truly hope that we have resolved the issues raised and would greatly appreciate it if you could let us know whether your concerns have been adequately addressed. If you have any further questions or would like additional clarifications, we would be more than happy to continue the discussion.
>
> Thank you once again for your valuable time and feedback. We look forward to hearing from you.
>
> Warm regards,
>
> All Anonymous Authors

---

### Official Review · Reviewer_c5w7 · 2024-11-06

**Soundness:** 3
**Presentation:** 3
**Contribution:** 3
**Rating:** 8
**Confidence:** 2

**Summary:**

The paper has two main contributions:
Firstly the authors conduct analysis on the theoretical properties of GaLore from the PL* condition perspective, which suggests that GaLore, compared with LoRA, has a larger region that satisfies the PL* condition, which provides faster convergence. Additionally, compared with full model training, the projected optimization approach suffers less from spurious minima issue, making it easier for optimization. These two arguments provide theoretical insights on the advantages of GaLore.
Then, the authors proposed a new method: GaRare, which uses random projection rather than SVD to find the low dimensional space, which improves memory efficiency in that the projection matrix no longer needs to be stored during training. GaRare, with lower memory cost, has the same theoretical properties of GaLore, as argued by the authors, and indeed shows performance comparable to GaLore on several empirical evaluation tasks, ranging from small-scale MLP-CIFAR10 experiments to LLM pre-training experiments.

Overall I find the paper well written, technically sound, and emprical results sufficiently supporting the claims.

**Strengths:**

- Parameter efficient learning / fine-tuning is a heated topic, lots of new methods are proposed but it is unclear why some methods are better / worse. The submission studies the advantages of GaLore and drawbacks of LoRA from an optimization perspective, providing useful insights and theoretical framework for understanding the PEFT methods.

- Learning rate swipe is conducted, making the emprical results more convincing!

- Low-bit training setting is considered, and the proposed method still shows performance competitive with GaLore, demonstrating the practical usefulness of the method.

**Weaknesses:**

- There exists couples of gap between the theoretical analysis's setting and the actual experiments: 1. The theory suggests a squared loss should be used but all experiments used cross entropy loss; 2. The theoretical analysis is limited to MLP.

- Same color is used to represent different ranks and optimization methods (Figure 1. and Figure 2.), which could be confusing. The authors could consider using different colormaps for these two figures.

- It is true that the random projection matrix does not need to be stored, but I think they still need to be materialized during forward propagation and would still cost O(mr) memory overhead if I understand correctly.

- (Minor) The proposed method uses random projection, which could introduce additional sensitivity to random seed in experiments.

**Questions:**

- The theoretical analysis focuses on train-from-scratch setting, is there any difference between training from scratch v.s. fine-tuning setting?

- How much extra runtime overhead would the random matrix generation cost in practice?

- How much difference would the "mr" term make in practice (Table 1, memory overhead of GaRare v.s. GaLore)? Since m is the output size, which would be pretty small in practice, e.g. 10 for cifar-10, and is much smaller than n, the input size, it would be nice if the authors could provide some example values of "m" and "r" in the caption of the table to better illustrate the differences.

---

> ### Author Response · Authors · 2024-11-23
> **Part 1**
>
> We thank the reviewer for their constructive feedback. Below, we address each of the concerns and questions raised:
>
> 1. **The theoretical analysis is limited to square loss and MLPs.**
>
> **Answer**: We believe that considering simplified settings, such as single architectures (MLPs) or loss functions (MSE), has already made significant contributions to the theoretical understanding of neural networks. Many influential works in the literature [1,2,3] also focus on simpler models and loss functions to provide rigorous insights into the optimization dynamics of neural networks. By adopting this approach, we aim to build a foundational framework that can guide future theoretical extensions and practical applications.
>
> Despite these simplifications, the insights gained from our analysis have practical implications beyond the specific settings studied:
>
> - For example, the GaRare algorithm, which was inspired by the theoretical insights presented in our paper, demonstrates strong performance on advanced models such as LLaMA and RoBERTa, despite these models are transformers and using cross-entropy loss in their training. This alignment between theoretical predictions and empirical results highlights the broader applicability of our framework and its utility in guiding algorithm design for large-scale, real-world models.
>
> - By focusing on MLPs and MSE loss, our work addresses the fundamental interactions between low-rank parameterizations, optimization landscapes, and memory efficiency. These insights are not limited to MLPs or MSE loss but serve as a starting point for analyzing more complex architectures and loss functions in future work.
>
> 2. **Same color is used to represent different ranks and optimization methods.**
>
> **Answer**: Thank you for your advice. We have modified this in our revised paper.
>
> 3. **The random projection matrix still need to be materialized during forward propagation and would still cost O(mr) memory overhead.**
>
> **Answer**:
> - During model inference and gradient backpropagation, the projection matrix is not required, so there is no additional memory overhead of \(O(mr)\) during these processes.
> - During the optimization step, the optimizer performs updates layer by layer rather than computing updates for all parameters simultaneously in GaLore and GaRare. Therefore, when computing the parameters for each layer, GaRare deletes the projection matrix corresponding to the previous layer's parameters. The projection matrix for an individual layer is very small and can be considered negligible in terms of memory usage during computation.
> - GaLore achieves such low memory consumption precisely also because of the layer-by-layer computation. Otherwise, during the SVD process, in addition to the projection matrix, an additional orthogonal matrix and a singular value vector would be generated, requiring an additional memory cost of \(O(nr)\). Because SVD is also performed layer by layer, the memory occupied by these additional matrices becomes negligible.
> - The layer-by-layer computation does not introduce a significant computational burden. As demonstrated in the runtime comparisons provided in the general response, the execution time of GaLore and GaRare is comparable to that of the full-rank standard Adam optimizer. This demonstrates that the optimization of memory usage does not come at the expense of runtime efficiency.
>
> 4. **The proposed method uses random projection, which could introduce additional sensitivity to random seed in experiments.**
>
> **Answer**: The impact of random projection on performance is minimal, as evidenced by the results of the second part of general response.
>
> 5. **The theoretical analysis focuses on train-from-scratch setting, is there any difference between training from scratch v.s. fine-tuning setting?**
>
> **Answer**: Compared to the train-from-scratch setting, fine-tuning involves a less challenging task. As a result, the impact of the optimization landscape is less significant. This can be observed by comparing the experimental results under these two settings: the advantages of GaLore and GaRare over LoRA are more evident in the train-from-scratch setting.
>
> 6. **How much extra runtime overhead would the random matrix generation cost in practice?**
>
> **Answer**: Nearly no extra runtime, as shown in the first part of General response.

---

> ### Author Response · Authors · 2024-11-23
> **Part 2**
>
> 7. **How much difference would the "mr" term make in practice?**
>
> **Answer**:
> - We apologize for the abuse of notation here. In Section 5.2, $m$ does not represent the output size but rather the number of rows of a specific parameter matrix. We have modified the notation in our revised version, replacing $m$ and $n$ with $p$ and $q$. The table shows the memory usage of weights and optimizer states for different algorithms applied to a particular parameter matrix. For each parameter matrix in the model, we have the results shown in the table. Therefore, when $m$ is large, meaning the network width is large, $mr$ becomes more significant.
> - From the results in Table 2 and Table 3, we can see that as the network structure increases, the advantages of GaRare also grow. In models with 100 billion parameters or more, GaRare achieves a more significant advantage over GaLore. Taking the LLaMA 1B model as an example, we set $r = 512$. The $m$ of each attention layer is 2048, and the $m$ of the MLP layers is either 2048 or 5461. The total parameter count of the 1B model is approximately 1227.05M, with the parameters included in "mr" accounting for about 248.00M.
>
> ### Reference
>
> [1] Lederer J. No spurious local minima: on the optimization landscapes of wide and deep neural networks[J]. 2020.
>
> [2] Liu C, Zhu L, Belkin M. Loss landscapes and optimization in over-parameterized non-linear systems and neural networks[J]. Applied and Computational Harmonic Analysis, 2022, 59: 85-116.
>
> [3] Liu C, Zhu L, Belkin M. On the linearity of large non-linear models: when and why the tangent kernel is constant[J]. Advances in Neural Information Processing Systems, 2020, 33: 15954-15964.

---

> ### Comment · Reviewer_c5w7 · 2024-11-23
> **Reviewer feedback**
>
> I would like to thank the author for the responses. I appreciate the authors for taking into account the comments and changing the color of figure lines as well as resolving the notation abuse of $mr$.
>
> > In models with 100 billion parameters or more, GaRare achieves a more significant advantage over GaLore
>
> Should it be with 100 **millions** parameters or more?
>
> I took a look at the updated manuscript, especially section 5.2, i.e. the discussion on memory usage of GaRare v.s. GaLore, I think I understand the advantages here after some careful read, however I believe the presentation has some large room for improvement (I do understand the current form could be written in a rush due to the tight timeline, so the authors could consider improving the presentation in future revision):
> - The inline table.1 makes the text hard to read
> - It would be nice if the authors could, e.g. have multiple \paragraph, or bulletpoints, and seperately discussing, e.g. forward overhead / backward overhead / optimizer state overhead, in which way the presentation would be much cleaner.
>
>
>
> I also took a look at the general response:
>
>
> Why does GaRare take more time than Full-Rank at 350M model?
>
>
> Regarding the ablation study on randomness, it is hard to directly tell if the GaRare is sensitive to random seed by looking at a standard deviation without having any references. I believe a better comparison would be to, e.g. also report the standard deviation of other methods, where random seed certainly also have effects through, e.g. initialization and dataset shuffling, then readers can have a better understanding of whether GaRare's variation is large or of similar magnitude with other methods.

---

> ### Author Response · Authors · 2024-11-25
> **Authors' feedback**
>
> We sincerely thank the reviewer for the detailed feedback and constructive suggestions. Below, we address the points raised and outline the corresponding updates made to the manuscript:
>
> 1. **Should it be 100 million or more?**
>
>    **Answer**: Yes, thank you for catching this error. We apologize for the oversight.
>
> 2. **Suggestions on the presentation.**
>
>    **Answer**: Thank you for the insightful feedback regarding the readability of Section 5.2. Based on your suggestions, we have restructured the section to improve clarity. Due to space limitations in the main text, we now provide a concise analysis in Section 5.2 while moving the detailed analysis to the appendix. We believe this adjustment enhances both readability and depth without sacrificing content accessibility.
>
> 3. **Time Overhead of GaRare at 350M Scale.**
>
>    **Answer**: While our experimental results show that GaRare incurs slightly higher computational time compared to the Full-Rank method at the 350M scale, the difference is within a comparable range. GaRare’s primary advantage lies in its substantial memory usage reduction, which is critical for scaling models efficiently. The observed time overhead is likely due to additional computations intrinsic to the GaRare methodology, as well as minor communication overhead between machine nodes during distributed processing. These factors are relatively insignificant and do not substantially affect overall performance.
>
> 4. **Should also report the standard deviation of other methods.**
>
>    **Answer**: Thank you for emphasizing the importance of providing the standard deviation of other methods. In response, we have included the standard deviation of Full-Rank training on the 60M and 130M LLaMA models for pretraining tasks, as well as on the RoBERTa-base model for fine-tuning tasks. These results highlight GaRare's stability compared to other methods. Due to time constraints, we were unable to include results for additional models and methods. However, we will address this in the final version to provide a more comprehensive analysis.
>
> |                | 60M  | 130M |
> |----------------|------|------|
> | Full-Rank      | 34.67$\pm$0.86| 24.87$\pm$0.18|
> | LoRA           | 34.99| 33.92|
> | FLoRA          | 36.97| 30.22|
> | GaLore         | 34.88| **25.36**|
> | GaRare         | **34.33±0.09**| **25.49±0.20**|
>
> |                | Full-Rank | GaRare    | GaLore    | FLoRA     | LoRA      |
> |----------------|-----------|-----------|-----------|-----------|-----------|
> | CoLA           | 62.5$\pm$0.2      | 61.0±0.1  | 60.4      | 59.0      | **61.4**      |
> | STS-B          | 90.8$\pm$0.1      | 90.3±0.0  | **90.7**      | 89.9      | 90.6      |
> | MRPC           | 91.2$\pm$0.0      | 91.4±0.2  | **92.3**      | 88.5      | 91.1      |
> | RTE            | 79.1$\pm$0.2      | **79.2±0.3**  | **79.4**      | 76.5      | 78.7      |
> | SST2           | 94.9$\pm$0.5      | **94.2±0.2**  | **94.0**      | 93.8      | 92.9      |
> | MNLI           | 87.2$\pm$0.1      | **87.0±0.2**  | **87.0**      | 86.6      | 86.8      |
> | QNLI           | 92.3$\pm$0.0      | **92.1±0.2**  | **92.2**      | 91.9      | **92.2**      |
> | QQP            | 92.3$\pm$0.0      | 90.9±0.3  | 91.1      | 90.9      | **91.3**      |
>
> 5. **Random seed certainly also have effects through, e.g. initialization and dataset shuffling.**
>
> **Answer**: We sincerely appreciate the reviewer’s insightful suggestion regarding the effects of random seeds on initialization and dataset shuffling. This is indeed an important consideration for providing a clearer understanding of GaRare's robustness relative to other methods.
>
> Due to time constraints, we were unable to include this comparison in the current version. However, we fully acknowledge its importance and plan to address this in the final version.

---

> > ### Comment · Reviewer_c5w7 · 2024-11-25
> > **Reviewer response**
> >
> > >  Based on your suggestions, we have restructured the section to improve clarity
> >
> > Thanks for the modification, I took a look at the appendix, and indeed find it much clearer!
> >
> > > The observed time overhead is likely due to additional computations intrinsic to the GaRare methodology, as well as minor communication overhead between machine nodes during distributed processing
> >
> > This makes sense! However I would suggest the authors to actually include such discussion into, e.g. the table/figure caption, together with something like "..., however it is worth noting that the primary advantage lies in its substantial memory usage reduction."
> >
> > > standard deviation of other methods.
> >
> > Thanks for the new results. I think the results are convincing about the sensitivity: GaRare's standard deviation looks roughly the same scale as full-rank. However one question here is that why is full-rank 60M's standard deviation so large? It would nice if the authors could have some explanation for this abnormal number.
> >
> > Nevertheless, I think this is an interesting paper. The main "limitation" is that the analysis is limited to MLP, which is totally acceptable for me: It is pretty common for deep learning theory papers to perform the analysis on a simple model and empirically verify the insights on more sophisticated models. I think the value of the paper is that it serves as a first step toward a deeper understanding of all these PEFT methods, at least for me, if someone asks me why GaRare/GaLore could be better than LoRA, I will point them to this submission and tell them it could potentially be an optimization problem.
> >
> > Lastly, I have increased my score from 6-8. However I am also open to opinions from other reviewers.

---

> > > ### Author Response · Authors · 2024-11-26
> > > **Thank you**
> > >
> > > Thank you for recognizing our work and providing valuable suggestions that have greatly improved the clarity and quality of our paper. We truly appreciate the time and effort you dedicated to offering such thoughtful feedback.
> > >
> > > Regarding the larger standard deviation of the full-rank 60M model, we found that this is caused by one specific seed yielding significantly poorer results compared to others. If we exclude this seed, the standard deviation returns to a normal range. We suspect this anomaly may be related to issues with data shuffling or initialization specific to that seed. This observation aligns with your earlier suggestion to conduct separate analyses on the effects of data shuffling and initialization. We believe such analyses could provide deeper insights into this behavior, and we plan to explore this direction further in future work. For now, we have chosen to report the unfiltered results to maintain transparency. Additionally, we aim to expand the number of experimental runs to ensure that the reported results are more robust and statistically reliable.

---

### Official Review · Reviewer_9dCt · 2024-11-09

**Soundness:** 4
**Presentation:** 3
**Contribution:** 3
**Rating:** 8
**Confidence:** 3

**Summary:**

The paper provides a theoretical analysis of low-rank adaptation methods for large language models (LLMs), focusing on LoRA and GaLore. The authors demonstrate that GaLore benefits from a more favorable optimization landscape than LoRA and full-rank methods, characterized by fewer spurious local minima and a larger region satisfying the PL* condition. Building on these insights, they introduce GaRare, a novel method that improves upon GaLore by employing gradient random projection to reduce computational overhead. Empirical results show that GaRare performs strongly in pre-training and fine-tuning tasks, offering an efficient approach to large-scale model adaptation.

**Strengths:**

+ I believe the paper is a great application of the PL* condition to fine-tuning LLMS which is of significant interest.
+ The theoretical analysis is rigorous and well-founded.
+ The paper contributes a valuable tool by introducing GaRare and demonstrating its practical efficiency.

**Weaknesses:**

- Some sections of the paper, especially those detailing the theoretical results could be more accessible--understandable given the page limit.

**Questions:**

N/A.

---

> ### Author Response · Authors · 2024-11-23
>
> Thank you for your recognition of our work and for pointing out the need for greater accessibility in the theoretical sections. We have included more explanations to clarify the concepts, motivations and insights behind the theoretical findings. This will help readers understand the main ideas and implications.

---

### Official Review · Reviewer_Nysi · 2024-11-09

**Soundness:** 3
**Presentation:** 3
**Contribution:** 3
**Rating:** 6
**Confidence:** 4

**Summary:**

This paper builds on previous work (LoRA and GaLore) that uses low-rank adaptation to reduce the memory footprint of LLM training. This paper provides an analysis of GaLore, which compresses the gradient matrix, showing it has fewer problematic local minima and better convergence properties. Based on this analysis, they propose a new method called GaRare that improves GaLore further by using random projections to reduce computational overhead. GaRare achieves reasonable better performance on both pre-training LLMs and fine-tuning them for specific tasks.

**Strengths:**

This paper provides a sound theoretical analysis that reveals the optimization landscapes of different adaptation methods for training Large Language Models (LLMs). A key strength of this paper is the in-depth examination of the parameter spaces where the PL∗ condition is satisfied, revealing that GaLore and full-rank methods have larger favorable regions compared to LoRA. This theoretical foundation explains the superior convergence rates and performance of GaLore by demonstrating its lower occurrence of spurious local minima, which often hinder optimization in training processes. Such insights into the conditions that facilitate faster and more reliable convergence are invaluable for developing more efficient LLM training techniques.

**Weaknesses:**

Limited Experiment Setup: The experimental validation of the proposed methods is confined to modifications of the MLP layer, whereas practical applications of LoRA predominantly target the attention layers within LLMs. This discrepancy raises concerns about the generalizability and practical relevance of the findings.

Marginal Performance Improvements: The performance improvements reported on the GLUE benchmark for GaRare, compared to the LoRA baseline and GaLore, are marginal. This minimal gain questions the practical value and impact of the proposed method in real-world application. Such incremental improvements might not justify the complexity or the computational costs associated with implementing GaRare.

**Questions:**

1) why we switch to RoBERTa for fine-tuning experiment? while the pretraining is performed on LLaMa? It would be more convincing and interesting to see FT results on LLaMa, which is the widely applied in practice.

2) Another benefit of LoRA is its application in low data regime, I wonder what is the results of the proposed methods at different training dataset size.

---

> ### Author Response · Authors · 2024-11-23
>
> We thank the reviewer for their constructive feedback. Below, we address each of the concerns and questions raised:
>
> 1. **The experimental validation of the proposed methods is confined to modifications of the MLP layer.**
>
> **Answer**:
> - This section of validation experiments is primarily designed to demonstrate the correctness of our theory, which is based on MLPs. Therefore, it is more appropriate to validate the theory on MLPs to get rid of the influence of complex network structure.
>
> - Although our analysis does not explicitly involve network structures like transformers, the experimental results in various aspects suggest that our theory also provides guidance for networks like transformers. Besides the general result that GaLore outperforms LoRA [1], [1] also found that when the rank increases to 2048, GaLore's performance surpasses that of the full-rank method. This aligns with the conclusion mentioned in our discussion in Section 4.3, where GaLore can achieve better performance than full-rank training when expressivity is not the bottleneck. This is also consistent with our experimental results on MLPs. The conclusion we reached that GaRare and GaLore exhibit similar performance is also reflected in the experiments. Thus, the conclusions we drew from MLPs can indeed generalize to transformers.
>
> - Moreover, our work can be regarded as a foundation for more refined theoretical studies in the future. Starting with the simplest MLPs, we verified its correctness, paving the way for further analysis of network structures like transformers, which could inspire the development of more effective algorithms.
>
> 2. **Marginal Performance Improvements.**
>
> **Answer**:
>
> - Our new algorithm, GaRare, does not offer advantages over GaLore in terms of performance improvements; rather, its primary benefit lies in reduced memory usage. Theoretically, GaRare should not outperform GaLore in terms of performance. From a practical perspective, as the scale of models continues to grow, memory consumption has become a widely recognized issue in large-scale model training.
>
> - Referring to the memory results presented in Table 2 and Table 3, GaRare demonstrates consistent memory advantages compared to GaLore. Furthermore, this advantage becomes more pronounced as the model size increases. For even larger models, such as those with 100 billion parameters, this benefit is expected to be even more significant. Therefore, our algorithm is more practical and applicable for larger-scale problems.
>
> 3. **Computational Costs of GaRare.**
>
> **Answer**:
> GaRare does not introduce higher complexity compared to GaLore and full-rank training. For more details, please refer to the results in the first part of "General Response" section.
>
> 4. **why we switch to RoBERTa for fine-tuning experiment?**
>
> **Answer**:
> - We followed the settings from the GaLore [1] to ensure a better comparison with GaLore. The reason for this setup is that the RoBERTa model is more suitable for classification tasks, while LLaMA is better suited for generative tasks.
>
> - Using different models for different tasks is a common practice in the field. For instance, LoRA [2] used RoBERTa on GLUE benchmark, while used GPT on E2E NLG Challenge, FLoRA [3] used T5 for XSum, GPT for IWSLT17.
>
> 5. **Another benefit of LoRA is its application in low data regime, I wonder what is the results of the proposed methods at different training dataset size.**
>
> **Answer**:
> We report the results of experiments in Section 6.2 when the data size is 1\% and 10\%. The advantages of GaLore and GaRare in low data regime become evident when data size reduces to 1\%. However, in this setting, the advantage of LoRA in the low-data regime is overshadowed by issues related to the optimization landscape, making it less effective than full-rank training.
>
> |                | 60M  | 130M | 350M | 1B   |
> |--------|------|------|------|------|
> | Full-Rank      | 63.43| **42.10**| 29.50| 26.98|
> | LoRA      | 83.15| 73.22| 49.51| 40.13|
> | FLoRA       | 70.43| 66.22| 42.62| 35.18|
> | GaLore         | **57.83**| 45.63| **28.79**| **26.81**|
> | GaRare         | 59.83| 48.69| 30.82| 28.30|
> | Training Tokens| 0.11B| 0.22B| 0.64B| 1.31B|
>
> |                | 60M   | 130M  | 350M  | 1B    |
> |-------|-------|-------|-------|-------|
> | Full-Rank | **441.42**| 468.72| 368.81| 237.45|
> | LoRA    | 841.56| 511.35| 502.17| 213.52|
> | FLoRA   | 541.17| 367.18| 158.73| 155.54|
> | GaLore    | 502.58| **317.61**| **139.29**| **145.70**|
> | GaRare | 445.74| 319.02| 162.53| 149.91|
> | Training Tokens| 0.01B | 0.02B | 0.06B | 0.13B |
>
> [1] Zhao J, Zhang Z, Chen B, et al. Galore: Memory-efficient LLM training by gradient low-rank projection[J]. arXiv preprint arXiv:2403.03507, 2024.
>
> [2] Hu E J, Shen Y, Wallis P, et al. LoRA: Low-rank adaptation of large language models[J]. arXiv preprint arXiv:2106.09685, 2021.
>
> [3] Hao Y, Cao Y, Mou L. FLoRA: Low-Rank Adapters Are Secretly Gradient Compressors[J]. arXiv preprint arXiv:2402.03293, 2024.

---

### Author Response · Authors · 2024-11-23
**General Response**

We sincerely thank all the reviewers for their valuable feedback and constructive comments on our manuscript. Your insights have greatly helped us improve the quality and clarity of the paper. Below, we address the general questions raised by the reviewers:

1. **The computational cost of GaRare.**

**Answer**: GaRare does not increase computational complexity compared to GaLore. Although GaRare requires continuously generating random matrices, this overhead is minimal. Moreover, it reduces the need for SVD computations compared to GaLore. Overall, it may even result in reduced training time. The specific training times are reported in the table below.

|                | 60M (1gpu) | 130M (2gpu) | 350M (2gpu) | 1B (4gpu) |
|----------------|------------|-------------|-------------|-----------|
| Full-Rank      | 1.98h      | 6.30h       | 26.47h      | 133.54h   |
| LoRA           | 1.93h      | 5.73h       | 24.87h      | 127.14h   |
| FLoRA          | 1.97h      | 6.00h       | 26.44h      | 129.88h   |
| GaLore         | 1.92h      | 6.07h       | 26.98h      | 131.55h   |
| GaRare         | 1.97h      | 5.96h       | 26.50h      | 130.00h   |
| $r / d_{\text{model}}$ | 128 / 256  | 256 / 768   | 256 / 1024  | 512 / 2048|
| Training Tokens| 1.1B       | 2.2B        | 6.4B        | 13.1B     |

2. **The randomness of GaRare.**

**Answer**: According to our theoretical analysis, as established in Lemma 5.1 and Corollary 5.2, the performance of GaRare is highly robust to the randomness of the random projection matrix, particularly when it is sampled from Gaussian distributions. These results suggest that the randomness of the random matrix has minimal impact on the algorithm’s overall performance.

In the original submission, we did not report results across multiple random seeds due to this theoretical robustness. To further validate this claim, we have now included results averaged over three random seeds. For experiments on LLMs, the mean and standard deviation are provided directly in the tables below. For the CIFAR datasets, these results are detailed in Section 6.1 of the revised manuscript. Across all cases, the results demonstrate small standard deviations, confirming our theoretical predictions and showing that GaRare's performance is stable and consistent across different random seeds.

We hope the additional analysis addresses the reviewer’s concern and strengthens the evidence for the robustness of GaRare.

### Pretraining LLaMA on C4 dataset.
|                | 60M  | 130M | 350M | 1B   |
|----------------|------|------|------|------|
| Full-Rank      | 34.06| 25.08| 18.80| 15.56|
| LoRA           | 34.99| 33.92| 25.58| 19.21|
| FLoRA          | 36.97| 30.22| 22.67| 20.22|
| GaLore         | 34.88| **25.36**| **18.95**| **15.64**|
| GaRare         | **34.33±0.09**| **25.49±0.20**| 19.24±0.07| **15.69±0.14**|
| $r / d_{\text{model}}$ | 128 / 256 | 256 / 768 | 256 / 1024 | 512 / 2048 |
| Training Tokens| 1.1B | 2.2B | 6.4B | 13.1B |

### Finetuning RoBERTa-Base and RoBERTa-Large on GLUE benchmark.

RoBERTa-Base
|                | Full-Rank | GaRare    | GaLore    | FLoRA     | LoRA      |
|----------------|-----------|-----------|-----------|-----------|-----------|
| CoLA           | 62.2      | 61.0±0.1  | 60.4      | 59.0      | **61.4**      |
| STS-B          | 90.9      | 90.3±0.0  | **90.7**      | 89.9      | 90.6      |
| MRPC           | 91.3      | 91.4±0.2  | **92.3**      | 88.5      | 91.1      |
| RTE            | 79.4      | **79.2±0.3**  | **79.4**      | 76.5      | 78.7      |
| SST2           | 94.6      | **94.2±0.2**  | **94.0**      | 93.8      | 92.9      |
| MNLI           | 87.2      | **87.0±0.2**  | **87.0**      | 86.6      | 86.8      |
| QNLI           | 92.3      | **92.1±0.2**  | **92.2**      | 91.9      | **92.2**      |
| QQP            | 92.3      | 90.9±0.3  | 91.1      | 90.9      | **91.3**      |
| Avg            | 86.3      | 85.8      | **85.9**      | 84.6      | 85.6      |

RoBERTa-Large
|                | Full-Rank | GaRare    | GaLore    | FLoRA     | LoRA      |
|----------------|-----------|-----------|-----------|-----------|-----------|
| CoLA           | 68.0      | 67.6±0.4  | **68.3**      | 65.5      | 68.2      |
| STS-B          | 91.5      | 92.3±0.0  | 92.5      | 92.5      | **92.6**      |
| MRPC           | 90.9      | **91.7±0.1**  | **91.7**      | 89.3      | 90.9      |
| RTE            | 86.6      | **87.2±0.3**  | 87.0      | 83.0      | **87.4**      |
| SST2           | 96.4      | **96.0±0.3**  | 96.1      | 96.0      | **96.2**      |
| MNLI           | 90.2      | **91.2±0.4**  | 90.8      | 90.4      | 90.6      |
| QNLI           | 94.7      | 94.5±0.1  | **95.7**      | 94.0      | 94.9      |
| QQP            | 92.2      | **91.9±0.2**  | **91.9**      | 91.5      | 91.5      |
| Avg            | 88.8      | 89.1      | **89.3**      | 87.8      | 89.0      |

---

### Meta-Review · Area_Chair_45Pr · 2024-12-23

**Metareview:**

Summary of Paper’s Contributions:
This paper provides a theoretical analysis of low-rank adaptation techniques (LoRA and GaLore) for large language model (LLM) training from an optimization landscape perspective. It shows that GaLore exhibits fewer spurious local minima and a larger region satisfying the Polyak-Łojasiewicz (PL*) condition, providing theoretical grounding for its empirical advantages. Building on these insights, the authors introduce GaRare, which replaces SVD-based projection matrices with randomly generated matrices, thereby improving memory efficiency without sacrificing performance. The theoretical analysis, albeit done on MLPs with MSE loss, offers intuition that appears to carry over to larger transformer-based models and cross-entropy loss scenarios, as evidenced by additional experiments on LLaMA and RoBERTa.

Key Strengths:
	1.	Novel Theoretical Insight: The paper moves beyond expressivity arguments and provides a complementary perspective focusing on the optimization landscape. This fills a critical gap in understanding parameter-efficient fine-tuning methods.
	2.	Practical Algorithm (GaRare): The proposed GaRare method reduces memory usage while maintaining the favorable optimization properties of GaLore. In large-scale scenarios, memory constraints are increasingly important, making GaRare particularly appealing.
	3.	Empirical Validation: The authors support their theoretical claims with experiments on both smaller MLP-based tasks and large-scale LLM training. They demonstrate that GaRare can achieve comparable performance to GaLore and LoRA while offering better memory efficiency.
	4.	Robustness and Stability: The authors provide standard deviation results and analyze the effect of randomness in GaRare, showing that performance is stable across different random seeds.

**Additional Comments On Reviewer Discussion:**

Key Weaknesses and Concerns:
	1.	Scope of Theoretical Analysis: The theoretical results are restricted to MLPs trained with MSE loss. While this is not uncommon in theoretical papers, some reviewers found the direct relevance to LLMs and transformer architectures less immediate. The authors have argued that this is a first step, and the intuition generalizes, but a direct extension to transformers remains future work.
	2.	Incremental Nature of GaRare: GaRare can be seen as a memory-optimized variant of GaLore rather than introducing a fundamentally new theoretical framework. However, the authors stress that the main contribution is the theoretical understanding, which guided the development of GaRare.
	3.	Lack of Comparison With More Complex PEFT Methods: Some reviewers requested comparisons with newer methods (e.g., AdaLoRA, SoRA) and higher LoRA ranks. While additional experiments were added in rebuttals, a full exploration, especially involving methods combining low-rank and sparsity, was not fully addressed. The authors note that integrating sparsity into their theoretical framework is left for future work.
	4.	Complexity of Empirical Setups: Some points, such as providing more thorough runtime analyses or clarifying certain anomalies in reported standard deviations, were raised. The authors provided additional details and clarifications in their rebuttals.

Discussion Among Reviewers:
	•	Most reviewers acknowledged the value of the theoretical analysis, seeing it as a meaningful addition to the literature, especially given the relative paucity of theory explaining why certain low-rank methods outperform others.
	•	Concerns remained about the limited theoretical scope and how well the results generalize to transformer-based LLMs. However, the authors convincingly argue that MLPs are a common starting point for theoretical deep learning research, and the good empirical performance on LLMs indicates that the insights likely carry over.
	•	Reviewers appreciated that the authors conducted additional experiments (e.g., different LoRA ranks, reporting std. dev.) and clarified memory usage and runtime overhead. They also recognized the difficulty in thoroughly covering newer LoRA variants combining sparsity and low-rank methods within the paper’s current scope.

Final Recommendation:
This paper provides a meaningful theoretical lens on low-rank adaptation methods and introduces a more memory-efficient variant, GaRare, supported by both theory and experiments. While not all theoretical questions are fully resolved, and the analysis is conducted in a simplified setting, the contributions are substantial enough to warrant acceptance. The paper illuminates previously unexplored aspects of low-rank adaptation’s optimization landscape and offers practical insights useful for training increasingly large models under memory constraints. Future work can build on this theoretical foundation to handle more complex architectures and extended settings.

Verdict: Borderline Accept

In light of the constructive dialog and revisions provided by the authors, I believe the paper should be accepted. The strengths, especially in providing theoretical understanding and delivering a practically useful method, outweigh the concerns. The paper will be a valuable starting point for subsequent theoretical and practical advances in parameter-efficient adaptation of LLMs.

---

### Decision · Program_Chairs · 2025-01-22

Accept (Poster)